# Coevolving with the Other You: Fine-Tuning LLM with Sequential Cooperative Multi-Agent Reinforcement Learning

**Hao Ma**[1,2*]   **Tianyi Hu**[1,2*]   **Zhiqiang Pu**[1,2†]   **Boyin Liu**[3]
**Xiaolin Ai**[2]   **Yanyan Liang**[4]   **Min Chen**[2]
[1]School of Artificial Intelligence, University of Chinese Academy of Sciences
[2]Institute of Automation, Chinese Academy of Sciences
[3]Alibaba (China) Co., Ltd.
[4]Macau University of Science and Technology
`{mahao2021, hutianyi2021, zhiqiang.pu, xiaolin.ai, chenmin2020}@ia.ac.cn`
`liuboyin.lby@alibaba-inc.com`
`yyliang@must.edu.mo`

## Abstract

Reinforcement learning (RL) has emerged as a pivotal technique for fine-tuning large language models (LLMs) on specific tasks. However, prevailing RL fine-tuning methods predominantly rely on PPO and its variants. Though these algorithms are effective in general RL settings, they often exhibit suboptimal performance and vulnerability to distribution collapse when applied to the fine-tuning of LLMs. In this paper, we propose CORY, extending the RL fine-tuning of LLMs to a sequential cooperative multi-agent reinforcement learning framework, to leverage the inherent coevolution and emergent capabilities of multi-agent systems. In CORY, the LLM to be fine-tuned is initially duplicated into two autonomous agents: a pioneer and an observer. The pioneer generates responses based on queries, while the observer generates responses using both the queries and the pioneer's responses. The two agents are trained together. During training, the agents exchange roles periodically, fostering cooperation and coevolution between them. Experiments evaluate CORY's performance by fine-tuning GPT-2 and Llama-2 under subjective and objective reward functions on the IMDB Review and GSM8K datasets, respectively. Results show that CORY outperforms PPO in terms of policy optimality, resistance to distribution collapse, and training robustness, thereby underscoring its potential as a superior methodology for refining LLMs in real-world applications. The code can be found at: `https://github.com/Harry67Hu/CORY`.

## 1   Introduction

Large language models (LLMs) have achieved impressive success across diverse downstream tasks, including dialogue systems [Ouyang et al., 2022, Touvron et al., 2023], code generation [Roziere et al., 2023], and robotic control [Driess et al., 2023, Brohan et al., 2023]. However, as the capabilities of LLMs advance, the challenges associated with further performance gains become increasingly intricate. Fine-tuning LLMs for specific tasks presents a significant challenge, prompting recent exploration of LLM fine-tuning paradigm such as supervised fine-tuning (SFT) [Wu et al., 2021], reinforcement learning (RL) fine-tuning [Shojaee et al., 2023], and direct preference optimization

---

[*]These authors contributed equally to this work.

[†]Corresponding author: zhiqiang.pu@ia.ac.cn.

38th Conference on Neural Information Processing Systems (NeurIPS 2024).

(DPO) [Rafailov et al., 2024]. RL fine-tuning demonstrates promising potential for refining LLM. Compared to SFT, RL fine-tuning offers a more direct optimization path, aligning training with desired outcomes and potentially leading to better out-of-distribution performance [Kirk et al., 2023]. Compared to DPO, RL fine-tuning allows fine-tuning on rule-based reward functions without requiring preference data.

However, contemporary RL algorithms are not specifically designed for LLMs. When fine-tuning an LLM using these RL algorithms, they exhibit instability and vulnerability to distribution collapse, which means that the LLM is over-optimized and exhibits highly biased behavior [Zheng et al., 2023, Yang et al., 2024b]. From the perspective of RL, LLM fine-tuning has several challenges, including large discrete action space and sparse rewards. Taking the RL fine-tuning of Llama-2 [Touvron et al., 2023] as an example, the dimension of the action space of Llama-2 can reach to 32000, representing 32000 potential vocabulary choices. Moreover, the reward signal is received only after generating the complete response, which results in a sparse reward problem. The above challenges hinder the exploration in such a vast search space, causing the instability of popular algorithms like PPO [Schulman et al., 2017].

Cooperative multi-agent reinforcement learning (MARL) represents a paradigm shift in the field of artificial intelligence (AI), where multiple autonomous agents coevolve within a complex system, resulting in the emergence of new skills [Foerster, 2018, Yang and Wang, 2020, Oroojlooy and Hajinezhad, 2023, Zang et al., 2023]. Language is an outcome of such multi-agent coevolution. In a society, numerous individuals utilize language for communication. Languages develop through agent interactions and are shaped by societal and cultural influences. As languages progress, they influence and are influenced by these interactions [Cavalli-Sforza and Feldman, 1981, Duéñez-Guzmán et al., 2023]. Inspired by this, fine-tuning an LLM within a cooperative MARL framework might lead to the emergence of superior policies during coevolution.

In this paper, we propose a plug-and-play method named CORY, which extends the RL fine-tuning of LLMs to a sequential cooperative MARL framework. In CORY, the LLM to be fine-tuned is initially duplicated into two autonomous agents[3], assigned two roles respectively: a pioneer and an observer. There are two fundamental mechanisms in CORY to enable the coevolution of the two LLM agents. The first is knowledge transfer, where the pioneer generates a response according to a task query independently, and the observer generates response based on the query as well as the response from the pioneer. The second is role exchange, where the roles of the two LLM agents are exchanged periodically during training. The two agents share a collective reward, calculated as the sum of individual task rewards, and they are trained simultaneously with their respective samples. Ultimately, CORY acts as a form of bootstrapping, wherein the collaborative learning between LLMs enhances the effectiveness of RL fine-tuning. Notably, this approach remains algorithm-agnostic, offering flexibility for integration with various RL algorithms beyond PPO, while maintaining simplicity and compatibility with existing methods.

In the experimental evaluation, we systematically investigate the efficacy of our proposed method across two types of reward functions: subjective and objective. Subjective reward functions are models trained to align human preferences, while objective reward functions are pre-defined functions typically established by domain experts. For the assessment of subjective rewards, we leverage the IMDB review dataset [Tripathi et al., 2020], a well-established benchmark for sentiment analysis. Meanwhile, the evaluation of objective rewards is conducted using the GSM8K dataset [Cobbe et al., 2021a], which focuses on mathematical word problem reasoning. Experiment results indicate that CORY surpasses PPO regarding policy optimality, resilience to distribution collapse, and robustness during training, highlighting its potential as an advanced method for improving LLMs in practical applications.

## 2  Problem Formulation

To understand LLMs through the lens of RL, we present a sequential decision-making problem formulation for the next-token prediction in causal language models. The next-token prediction is precisely defined by the concept of language-augmented Markov decision process [Li et al., 2022], denoted as $\mathcal{M} =< \mathcal{V}, \mathcal{S}, \mathcal{A}, r, P, \gamma >$. Here, $\mathcal{V}$ represents a vocabulary of a language model,

---

[3]The "agents" here refer to individuals who make decisions and take actions in the context of reinforcement learning [Sutton and Barto, 2018].

encompassing all possible tokens. The $w \in \mathcal{V}$ represents a specific token within this vocabulary. The state space $\mathcal{S} \subset \mathcal{V}^M$, where $\mathcal{V}^M$ is the combination space of $M$ tokens. The action space $\mathcal{A} \subset \mathcal{V}^N$, where $\mathcal{V}^N$ is the combination space of $N$ tokens. $M$ and $N$ are the max token lengths for state and action, respectively. A state $s \in \mathcal{S}$ is a concatenation of token sequence $s = (w_1, w_2, \ldots, w_M)$. An action $a \in \mathcal{A}$ is the output of a causal language model, construed as a concatenation of token sequence $a = (w_1, w_2, \ldots, w_N)$. The states and actions are padded with pad token if the real length is less than the maximum length. The reward function $r : \mathcal{S} \times \mathcal{A} \to \mathbb{R}$ assigns a numerical score to a sequence of tokens, which can be considered as a typical sparse reward problem within the context of RL. The state transition function $P : \mathcal{S} \times \mathcal{V} \to \mathcal{S}$ describes a deterministic transition of states according to the auto-regressive paradigm. At each step, a predicted token is concatenated with the state of last step: $s_{i+1} = (s_i, w_{i+1}) = (s_0, w_{1:i+1})$, where $s_0$ denotes a tokenized user's input for a causal language model, and $w_{1:i} = (w_1, w_2, \ldots, w_i)$ denotes a token sequence up to the $i$-th token. Then, the token-level policy of a causal language model can be encapsulated within $\pi(w_i | s_0, w_{1:i-1})$. And the sentence-level policy is defined as a joint policy:

$$\pi(a|s_0) = \prod_{i=1}^{N} \pi(w_i | s_0, w_{1:i-1}). \tag{1}$$

The reward function $r(\cdot, \cdot)$ is related to a specific task (e.g., safety alignment [Liu, 2023, Ji et al., 2024], code generation [Shojaee et al., 2023, Liu et al., 2023]). A task reward is only obtained after $N$ steps of decision-making via token-level policy. Under such a sparse reward, RL is prone to over-optimisation, resulting in distributional collapse of the language model. To mitigate the risk of distributional collapse, it is common practice to incorporate token-level KL penalties into the reward function, which serves to constrain the deviation of the language model from its original distribution [Go et al., 2023, Zheng et al., 2023].

$$\hat{r}(s_i, w_i) = \begin{cases} -\eta KL(\pi_\theta(\cdot | s_0, w_{1:i-1}), \pi_0(\cdot | s_0, w_{1:i-1})) & i < N \\ r(s_0, a) - \eta KL(\pi_\theta(\cdot | s_0, w_{1:i-1}), \pi_0(\cdot | s_0, w_{1:i-1})) & i = N, \end{cases} \tag{2}$$

where $\eta$ is the KL coefficient, $\hat{r}(s_i, w_i)$ represents the token-level combined reward function. For each token, a KL penalty is imposed based on the KL divergence between current policy $\pi_\theta(\cdot | s_0, w_{1:i-1})$ and initial policy $\pi_0(\cdot | s_0, w_{1:i-1})$. Only after predicting the final token, does the reward model yield a task-specific reward $r(s_0, a)$.

## 3 Method

### 3.1 Coevolving with the Other You (CORY)

To extend the RL fine-tuning of LLMs to a cooperative MARL framework, the LLM to be fine-tuned in CORY is initially duplicated into two copies, each is treated as an autonomous agent. Then, two roles, a pioneer and an observer, are assigned to these two LLM agents. We design two fundamental mechanisms to facilitate the coevolution between the two agents. The first design is knowledge transfer. The LLMs asynchronously take action, with the pioneer transferring its response (action) to the observer. The observer then utilizes this information to guide its own decision. The second design is role exchange. Once the observer achieves a satisfactory performance, it exchanges roles with the pioneer. In the following, we provide a comprehensive description of each element, and the pipeline of our method is shown in Figure 1.

**Knowledge Transfer.** To enable collaboration between the two LLM agents for improved response generation, we introduce a knowledge transfer mechanism. Given a query denoted as $s_0$, the pioneer acts first and generates a response denoted as $a_1$. Subsequently, the observer receives both the original query $s_0$ and the pioneer's response $a_1$ to generate its own response $a_2$. This sequential interaction facilitates knowledge transfer, where the observer leverages the pioneer's output to guide its own generation process, potentially leading to a superior response due to the in-context learning capabilities of LLMs. The sentence-level policies of the pioneer and observer can be formulated as follows:

$$a_1 \sim \pi_{\text{pio}}(\cdot | s_0), \quad a_2 \sim \pi_{\text{obs}}(\cdot | s_0, a_1). \tag{3}$$

During the training process, the parameters of the pioneer and the observer are optimized separately through an RL algorithm such as PPO. A cooperative relationship exists between the two LLM agents.

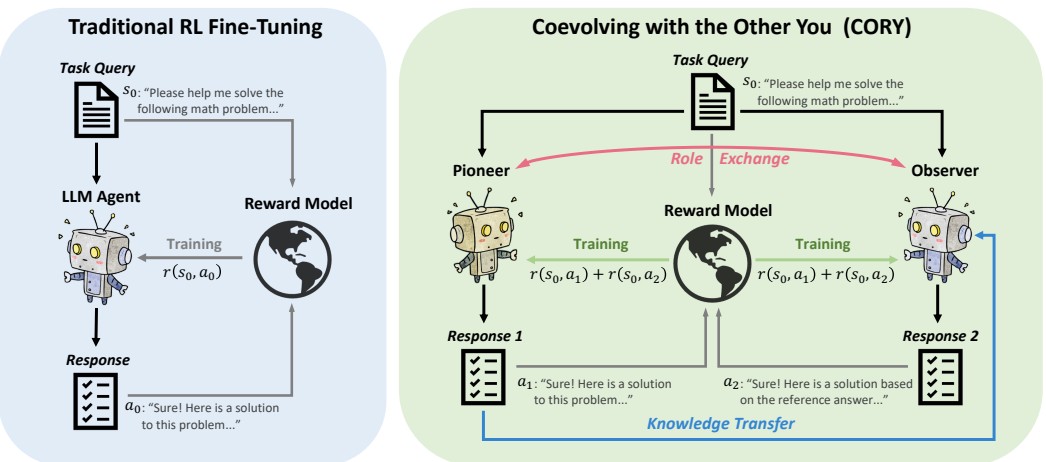

Figure 1: The framework of CORY. A traditional RL fine-tuning method can be simply extended to the CORY version with only three steps. First, duplicate the LLM into two LLM agents, one acting as a pioneer and the other as an observer; second, combine the task rewards of the two LLM agents to replace the original task reward; third, periodically exchange the roles of the two LLM agents during training. After training, either the LLM agent can perform the task independently.

To facilitate this collaboration, CORY employs a collective task reward, calculated as the sum of individual task rewards:

$$r_{\text{CORY}}(s_0, a_1, a_2) = r(s_0, a_1) + r(s_0, a_2), \tag{4}$$

which implies that both the pioneer and the observer receive rewards from each other's improvement. Following the form of Equation 2, we add $r_{\text{CORY}}$ and the KL penalty to construct a whole reward signal. Similar to Ni et al. [2022], we find that a partially correct reference can also be beneficial for the observer. Hence, it is not necessary for the pioneer to generate a high-quality response.

**Role Exchange.** During training, the observer may develop a prompt bias due to consistently receiving inputs in the form of $(s_0, a_1)$. This reliance on prompts that combine the original query with the pioneer's response, hinders the observer's ability to generate responses independently. To address this issue, we introduce a role exchange mechanism. This mechanism involves exchanging the roles of the pioneer and observer periodically during training:

$$\begin{aligned}
\pi_{\text{pio}}(\cdot|s_0) &= \pi_{\text{pio}}(\cdot|s_0; \theta_1), \quad \pi_{\text{obs}}(\cdot|s_0, a_1) = \pi_{\text{obs}}(\cdot|s_0, a_1; \theta_2), \text{ if } swap = False \\
\pi_{\text{pio}}(\cdot|s_0) &= \pi_{\text{pio}}(\cdot|s_0; \theta_2), \quad \pi_{\text{obs}}(\cdot|s_0, a_1) = \pi_{\text{obs}}(\cdot|s_0, a_1; \theta_1), \text{ if } swap = True,
\end{aligned} \tag{5}$$

where $swap$ is initialized as $False$, and reverse periodically. This exchange ensures that both the LLMs experience both roles (pioneer and observer) multiple times throughout the training process. Through this role exchange mechanism, they are forced to adapt to both prompt formats: $s_0$ alone and the combined format $(s_0, a_1)$. This allows us to use either LLM individually during inference. From a representational learning perspective, this role exchange mechanism encourages the LLMs to develop a unified representation for $s_0$ and $(s_0, a_1)$. This unified representation captures the essential information from the task query, regardless of the specific prompt format presented during training or inference.

These two key mechanisms in CORY act as a form of bootstrapping. The two LLM agents collaborate, with the observer potentially learning better policies by leveraging the pioneer's output. Role exchange ensures both the LLMs benefit from this collaborative learning, similar to cooperative learning among humans. Importantly, CORY is an algorithm-agnostic approach, meaning it can be theoretically compatible with various RL algorithms beyond PPO. Additionally, CORY offers the advantages of simplicity in implementation and seamless integration with existing frameworks, making it a plug-and-play solution. The derivation of the CORY's policy update can be found in Appendix B, and the detailed pseudocodes are provided in Appendix C.

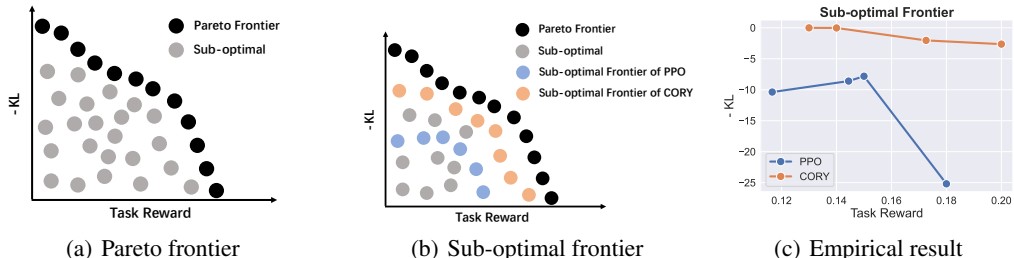

|                    |                         |                     |
|--------------------|-------------------------|---------------------|
| (a) Pareto frontier | (b) Sub-optimal frontier | (c) Empirical result |

Figure 2: The empirical demonstration of why CORY surpasses single-agent RL fine-tuning. In (c), the values of $\eta$ from left to right are 1e-5, 1e-4, 1e-3, and 1e-2.

## 3.2 Understanding CORY

Following the explanation of CORY in Section 3.1, this section provides an empirical demonstration of why the proposed method surpasses the single-agent RL fine-tuning method.

In fact, RL fine-tuning with KL penalty inherently formulates a multi-objective reinforcement learning problem. The LLM agent strives to concurrently maximize the task reward and minimize the KL divergence. Unfortunately, these two objectives may be in opposition to one another. This is because maximizing the task reward will inevitably lead to the output distribution deviating from the pre-trained model, resulting in an increase in KL divergence. Hence, the optimization process seeks a trade-off between the task reward and the KL divergence, ideally driving the policy towards a Pareto frontier [Ngatchou et al., 2005]. This frontier covers all achievable policies where no policy can improve on one objective without sacrificing performance on the other. Formally, the Pareto frontier can be defined as:

$$\mathcal{F} := \left\{ J_{\mathbf{r}}(\pi) \mid \pi \in \Pi \wedge \nexists \pi' \neq \pi : J_{\mathbf{r}}(\pi') \geq J_{\mathbf{r}}(\pi) \right\}, \tag{6}$$

where $J_{\mathbf{r}}(\pi) = \mathbb{E}_\pi[\sum_{t=0}^T \gamma \mathbf{r}(s_t, a_t)]$. $\mathbf{r}(s, a) \in \mathbb{R}^m$ is a vector-valued reward function and $\Pi$ denotes the set of all policies. Given a fixed reference vector $\boldsymbol{\omega} \in \boldsymbol{\Omega} \subseteq \mathbb{R}^m$, one could scalarize the multi-objective reward into a single objective by using the weighted sum $\boldsymbol{\omega}^T \mathbf{r}(s, a)$. Under this preference weighting, the ideal outcome for the policy is to converge to a point on the Pareto frontier, as illustrated by the black dots in Figure 2(a).

However, due to the inherent complexities of natural language, achieving perfect policy convergence to the Pareto frontier is often intractable. Nevertheless, by adjusting the preferences, these sub-optimal policies can still form a frontier as illustrated in Figure 2(b). For simplicity, we term it the sub-optimal frontier. Our hypothesis is that the sub-optimal frontier achieved by CORY lies closer to the true Pareto frontier compared to that achieved by single-agent RL method.

To verify this hypothesis, we fine-tune the Llama-2-7b-chat model on the grade school math 8K (GSM8K) dataset [Cobbe et al., 2021b] using both PPO and CORY. We measure the KL divergence and the task reward obtained by each policy after convergence. By adjusting the preference, i.e., $\eta$ in Equation 2, we are able to generate sub-optimal frontiers for both the methods, as illustrated in Figure 2(c). It is important to note that the Y-axis represents the negative KL divergence (larger values indicate better performance). As expected, the sub-optimal frontier achieved by CORY consistently outperforms that of PPO, empirically validating the hypothesis.

Our analysis through the lens of multi-objective RL offers valuable insights into the effectiveness of CORY. The knowledge transfer mechanism inherently addresses the optimization challenges faced by the observer. By leveraging the reference response provided by the pioneer, the observer actually experiences a guided optimization process. Such guided process can alleviate the optimization pressure on the task reward side, and prioritize improvement on the KL penalty side. However, since the observer's policy during training takes both the task query and the pioneer's response as inputs, the optimized policy is not the one we really want (we need the policy which only takes the task query as input), resulting in the prompt bias issue. The role exchange mechanism can effectively address this issue, and transfer the skills learned by the observer back to the pioneer, reducing the pioneer's optimization pressure. Notably, CORY demonstrates significantly better stability and robustness compared to single-agent RL method (See details in Section 4.2 and Appendix E.1). It consistently

achieves a lower KL divergence between the fine-tuned and pre-trained models while maintaining strong performance on the target task, signifying a better trade-off between the two objectives.

## 4 Experiments

This section systematically investigate the performance of CORY across two types of reward functions: subjective reward function and objective reward function. Subjective reward functions are reward models trained on data capturing human preferences. They essentially translate the human sentiment or judgment into a numerical reward signal that guides alignment. Objective reward functions are pre-defined rule-based functions, typically established by domain experts. This categorization reflects real-world scenarios where reward functions might be learned from human preferences or manually crafted by domain experts. Prompts used in experiments are detailed in Appendix A.2.

### 4.1 Subjective Rewards on IMDB Review

**Task Setup.** To evaluate our method under the subjective reward setting, we select the IMDB Review dataset [Tripathi et al., 2020]. This dataset contains 50K <text,label> pairs, with the training set and the test set each contains 25K pieces of data. The texts in the IMDB dataset are movie reviews, and the labels are the binary sentiment classification labels. The distilbert-imdb model[4] trained on the dataset is employed as the reward model. We fine-tune GPT2-Large (774M)[5] by using single-agent PPO (single-PPO) and CORY respectively. In addition, GPT2-XL (1.5B)[6] is fine-tuned by using single-PPO as an ablation on model size. In this task, we randomly sample text snippets from the IMDB dataset. The first 2 to 8 tokens (representing the beginning of the review) are retained as prompts for sentiment completion. The LLMs generate continuations that transform the prompts into positive sentiment comments. After that, the reward model evaluates the generated text to assign a sentiment score. The objective is to maximize the average sentiment score of the completed comments. Examples of this task are detailed in Appendix D.

In the experiments, each method undergoes 100 training iterations using a batch size of 256. For simplicity, GPT2-Large and GPT2-XL fine-tuned by single-PPO are termed as *PPO-GPT-2-l* and *PPO-GPT-2-xl*, respectively. GPT-2-Large that fine-tuned by CORY are referred to *CORY-LLM1* and *CORY-LLM2*, where the former one is the LLM that initialized as the pioneer, and the latter one is the LLM that initialized as the observer.

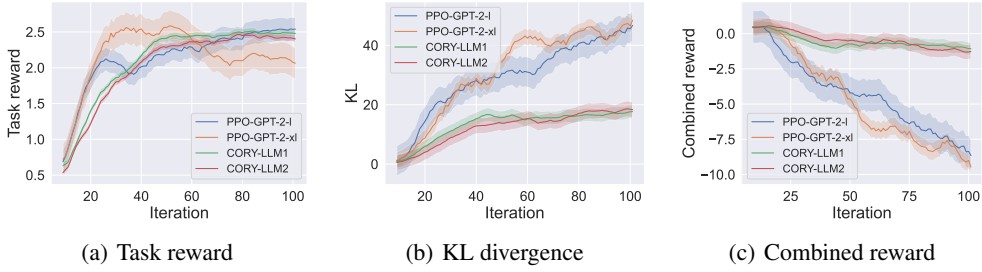

(a) Task reward  (b) KL divergence  (c) Combined reward

Figure 3: Training curves under subjective rewards on IMDB Review.

**Results and Analysis.** We monitor the training process by visualizing task reward, KL divergence, and a combined reward function that incorporates both the above objectives. Denoted as $r_c(s_0, a)$, the combined reward function can be expressed as $r_c(s_0, a) = r(s_0, a) + \eta * KL(s_0, \pi_\theta, \pi_0)$, where $r(s_0, a)$ and $KL(s_0, \pi_\theta, \pi_0)$ are the sentence-level task reward part and the KL penalty part, respectively. And the KL penalty part can be calculated as $KL(s_0, \pi_\theta, \pi_0) = \sum_{i=0,1,...,N} -KL(\pi_\theta(\cdot|s_0, w_{1:i-1}), \pi_0(\cdot|s_0, w_{1:i-1}))$.

It is important to note that, the actual reward used for training in CORY is not the combined reward. The actual training reward not only includes the KL penalty and the task reward from the target agent,

---

[4]https://huggingface.co/lvwerra/distilbert-imdb
[5]https://huggingface.co/openai-community/gpt2-large
[6]https://huggingface.co/openai-community/gpt2-xl

but also includes the task reward from the other agent. In fact, the combined reward $r_c(s_0, a)$ is the real overall objective that needs to be optimized, and can be aligned with the single-agent RL fine-tuning, making it easier to compare performance of all the methods.

The training curves of task reward, KL divergence, and the combined reward are illustrated in Figure 12. The results show that single-PPO and CORY achieve similar task reward levels after 100 training iterations. However, the curve of KL divergence related to single-PPO is significantly higher than that of CORY, reaching more than twice the level of CORY after all the training iterations. This indicates CORY's ability to achieve similar task reward levels with a smaller deviation from the pre-trained policy. Moreover, it can be observed that the curves of *CORY-LLM1* and *CORY-LLM2* are very close, indicating that the two LLM agents initially playing different roles finally achieve very similar performance levels at the end of the training. Consistent with the motivation of CORY, both the fine-tuned LLM agents can be used to finish tasks individually, which verifies the effectiveness of the bootstrapped learning and coevolution principles in CORY.

Finally, Figure 12(c) visually confirms CORY's advantage in combining the two objectives. The combined reward curve for CORY consistently rises, indicating its effectiveness in simultaneously improving task reward and minimizing KL divergence. Conversely, PPO's combined reward curve exhibits a decreasing trend, suggesting its struggle in balancing these objectives. Hyperparameters used for both single-PPO and CORY are detailed in Appendix A.1.

### 4.2 Objective Rewards on GSM8K

**Task Setup.** To evaluate our method under a rule-based objective reward function, we select the GSM8K task [Cobbe et al., 2021a]. GSM8K comprises 8.79K high-quality, linguistically diverse grade school math word problems, with 7.47K allocated for training and 1.32K for testing. For each question in the dataset, a response is obtained via LLM. The precise answer is extracted from the responses using a regular expression, typically the final set of numbers in the response. If the number in question matches the ground truth as recorded in the dataset, a reward of 1 is awarded. Conversely, if the number is incorrect, a reward of 0 is given. The Llama-2-7b-chat[7] model is selected as the pre-trained model. To reduce the training overhead, the model is quantised to 4-bit. For simplicity, the 4-bit Llama-2-7b-chat model fine-tuned with single-PPO is referred to as *PPO-Llama-2*. The copied models fine-tuned with CORY are referred to *CORY-LLM1* and *CORY-LLM2*, where the former is the LLM that initialized as the pioneer, and the latter is the LLM that initialized as the observer. Examples of this task are detailed in Appendix D.

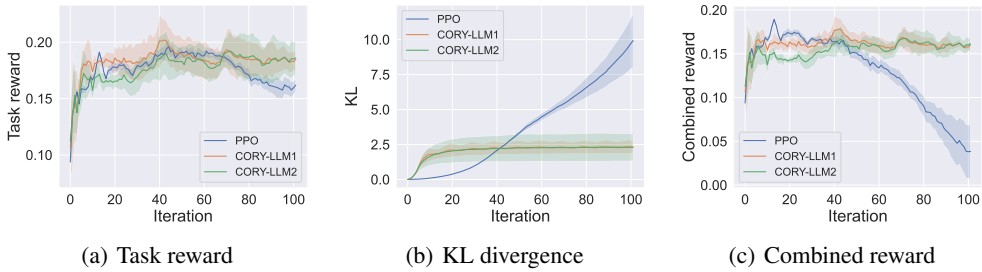

|  |  |  |
|:---:|:---:|:---:|
| (a) Task reward | (b) KL divergence | (c) Combined reward |

Figure 4: Training curves under objective rewards on GSM8K.

**Results and Analysis.** Similar to Section 4.1, We monitor the training process by visualizing task reward, KL divergence, and the combined reward. As shown in Figure 4, the jitter observed in all curves suggests the challenge posed by GSM8K. The vast exploration space presents inherent instability for the RL algorithms. As Figure 4(a) illustrates, the task reward curve of single-PPO peaks around 50 training iterations, followed by a decline. Single-PPO's KL divergence exhibits no convergence trend, reaching a maximum value during training (Figure 4(b)). The instability of single-PPO results the high KL divergence after 50 iterations, leading to a poor performance on combined reward (Figure 4(c)).

---

[7] https://huggingface.co/meta-llama/Llama-2-7b-chat-hf

In contrast, CORY demonstrates a significantly more stable task reward curve, consistently outperforming single-PPO. What's more, CORY achieves a considerably lower KL divergence compared to single-PPO, facilitating faster convergence. This characteristic is particularly valuable in the fine-tuning context, as it allows CORY to achieve similar or even better task rewards without significant modifications to the original parameter distributions.

Furthermore, the combined reward curves visually confirm CORY's superiority over single-PPO. CORY's ability to effectively balance the two objectives is reflected in its steadily increasing combined reward. Conversely, single-PPO's struggle with balancing the objectives manifest as a decreasing combined reward and training instability.

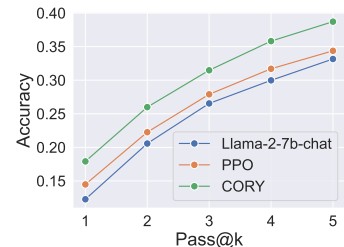

Figure 5: Evaluation results on GSM8K test dataset.

In addition, we conduct a comparative analysis of models fine-tuned with distinct methods and a pre-trained model on the GSM8K test set as shown in Figure 5. The evaluation metric utilized is $pass@k$, which generates $k$ corresponding repetitions for a sample and passes if at least one is correct. The test results demonstrate that the CORY fine-tuned 4bit Llama-2-chat-7b could achieve a $pass@1$ of $18\%$ on GSM8K test dataset.

## 4.3 Ablations

In ablation experiments, we ablate the influence of model size, knowledge transfer, and role exchange under the subjective reward setting on IMDB review dataset. For method names depicted in Figure 6, *REx* indicates role exchange, *KT* indicates knowledge transfer, *LLM1* and *LLM2* refer to LLMs who are initialized as the pioneer and the observer respectively.

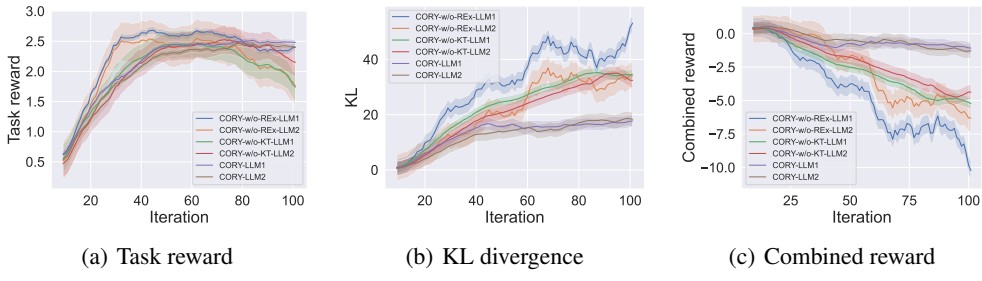

| (a) Task reward | (b) KL divergence | (c) Combined reward |

Figure 6: Training curves for ablations experiments.

**Ablation on Model Size.** Our method employs two models during training, with the total parameters trained being doubled in comparison to single-PPO. In order to ablate whether the enhancement of CORY is derived from the expansion of the model parameters, an additional fine-tuning of GPT2-XL (1.5B) with single-PPO is conducted on the IMDB dataset, which has twice the number of parameters as GPT2-Large. The results are presented in Figure 12. While the task reward of the model rapidly reaches its maximum value, the KL penalty part does not exhibit a notable improvement compared to GPT2-Large. The KL divergence continues to increase, leading to the collapse of the distribution.

**Ablation on Knowledge Transfer.** We maintain role exchange, and the two models still share a collective task reward (Equation 4), but disable knowledge transfer. This resembles PPO with individual queries as inputs. However, without the observability of the pioneer's outputs, this equivalent to adding noise to the PPO reward signal. Consequently, the task rewards become unstable, and the KL divergences are higher compared to CORY as shown in Figure 6. This highlights the importance of observability for framing RL fine-tuning as a true multi-agent cooperation problem.

**Ablation on Role Exchange.** We maintain knowledge transfer but disable role exchange. As evident from Figure 6, both LLMs achieve good task rewards, but their KL divergences are much higher than that of CORY. Notably, the observer LLM exhibits significantly lower KL divergence compared to the pioneer LLM. This observation highlights a fascinating phenomenon in cooperative learning:

by receiving the pioneer's response, the observer can effectively optimize the KL divergence. This suggests that the observer leverages the pioneer's exploration to refine its policy while maintaining good performance, potentially leading to a more stable learning process.

# 5 Related Work

The most related topic is reinforcement leanring from human feedback (RLHF). InstructGPT [Ouyang et al., 2022] fine-tunes GPT-3 like models [Brown et al., 2020] to enhance helpfulness by combining SFT with RL based on human preference dataset. Askell et al. [2021] trains a preference model for aligning the LLM with human values. It argues that ranked preference modeling proves to be the most effective training objective for distinguishing between desirable and undesirable LLM behaviors. Bai et al. [2022] incorporates an iterative online training mode where preference model and LLM are updated weekly using fresh human feedback data. Existing research acknowledges the inherent complexity, instability, and hyperparameter sensitivity of RLHF, particularly when employing PPO Zheng et al. [2023]. Several works have attempted to address these challenges by introducing max-entropy regularization [Wen et al., 2024], hyperparameter tuning [Zheng et al., 2023], and reward shaping [Yang et al., 2024a]. However, these methods does not show significant improvement over the vanilla PPO algorithm. This inspires us to explore alternative method from a different perspective that extent the RL fine-tuning of LLMs to a cooperative MARL problem.

Another related topic is MARL. Under the interaction relationship (cooperation, competition, mixed), multi-agent could spontaneously emerge complex and diverse policies, so as to solve the complex problems that single-agent reinforcement learning is difficult to solve. For example, in Kim et al. [2023], the RL based prompt tuning is decomposed into multi-agent joint tuning. The huge joint action space is equally split across agents, learning better and longer prompt. Such mechanisms have also been applied in the field of combinatorial optimization. The paper that is most similar to us on the architecture of agent training is Gao et al. [2023]. It proposes an asymmetric training symmetric execution framework to deal with the two-agent Stackelberg game Fang et al. [2021]. In the Stackelberg game, two agents make decisions asynchronously. The agent that makes the decision later can observe the former agent, but the former agent cannot observe the later agent. The training framework proposed by the authors is able to converge in Stackelberg equilibrium empirically. This inspires us to design the training framework for LLMs under a sequential cooperative setting.

# 6 Discussion

Experimental evidence suggests that CORY yields more stable and superior performance in RL fine-tuning. This can be attributed to our extension of single-agent RL fine-tuning into a cooperative MARL version. In this section, we delve into a discussion of how the multi-agent learning can benefit LLM fine-tuning. The primary benefit is that multi-agent learning encourages the coevolution of LLMs through collective living, social relationships and major evolutionary transitions [Duéñez-Guzmán et al., 2023]. This process generates a variety of new data, which further facilitates coevolution. This mechanism contributes to many breakthroughs in games AI, such as Go [Silver et al., 2016, 2017, Clark and Storkey, 2015], StarCraft II [Vinyals et al., 2019], and Diplomacy [Bakhtin et al., 2022].

In this paper, we investigate the application of cooperative MARL to address challenges in RL fine-tuning. Cooperative MARL fine-tuning appears to increase training robustness and prevent distribution collapse. While we concentrate on cooperation, competitive MARL, especially population-based methods, represents a promising direction for future research. These approaches create an auto-curriculum mechanism driven by a natural arms race, which propels agent learning and enables mastery of complex tasks. Besides the interaction paradigm, the scale of agents is crucial to emergence. While we examine a setting involving two LLMs, incorporating more LLMs in MARL fine-tuning is an intriguing prospect for future studies.

# 7 Conclusion

In this paper, we extend the RL fine-tuning of LLMs to a sequential cooperative MARL framework. To this end, we duplicate the pre-trained LLM into two LLM agents with different roles, and design two key mechanisms: knowledge transfer and role exchange. These mechanisms enable the two LLM

agents to learn collaboratively, and after the fine-tuning process, either the LLM agent can be chosen to perform the task independently. We also provide an in-depth analysis of RL fine-tune from the perspective of multi-objective RL, revealing the existence of a Pareto frontier between KL divergence and task reward. We empirically illustrate that CORY has an advantage over single-agent RL method in approaching the Pareto frontier. Experiment results indicate that CORY surpasses PPO regarding policy optimality, resilience to distribution collapse, and robustness during training, highlighting its potential as an advanced method for improving LLMs in practical applications.

## 8 Acknowledgement

This work was supported by the Strategic Priority Research Program of Chinese Academy of Science under Grant No. XDA27030204, the National Natural Science Foundation of China under Grant 62322316, the Beijing Nova Program under Grant 20220484077 and 20230484435.

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

# A  Implementation Details

The code repository we utilize is TRL[8]. Our experimentation employs 2 AMD EPYC 7773X CPUs and 8 NVIDIA A6000 GPUs (48GB each). Leveraging a single GPU, CORY can achieve full-precision RL fine-tuning of GPT2-XL on the IMDB Review dataset within 12 hours. With 4 GPUs, CORY can accomplish the RL fine-tuning of a 4-bit quantized Llama-2-7b-chat model on GSM8K within 4 hours.

## A.1  Hyperparameters

The hyperparameter settings for fine-tuning GPT2 followed the default configuration in TRL for the IMDB dataset, while the hyperparameter setting of Llama-2 primarily adhered to the guidelines provided by StackLlama. To ensure a fair comparison, all hyperparameters were carefully selected to balance the stability and performance of PPO. A grid search was conducted over $\alpha$ and $\eta$, with the sets $\alpha$ 1e-6, 1e-5, 1e-4 and $\eta$ 1e-3, 1e-2, 1e-1, 0.2, 0.3, respectively, to identify the hyperparameter that yielded the most stable training for PPO. Given CORY's robustness to hyperparameters (Appendix E.1), most PPO hyperparameters, except for the learning rate $\alpha$, were applied directly to CORY. For the GSM8K dataset, In the GSM8K dataset, we adjusted the learning rate $\alpha$ for CORY.

Table 1: Hyperparameters in IMDB Review

| Hyperparameter | PPO-GPT-2-l | PPO-GPT-2-xl | CORY |
|---|---|---|---|
| Learning Rate ($\alpha$) | 1.41e-5 | 1.41e-5 | 1.41e-5 |
| Epochs | 1 | 1 | 1 |
| PPO Epoch | 4 | 4 | 4 |
| Batch Size | 256 | 256 | 256 |
| Mini Batch Size | 256 | 256 | 256 |
| Gradient Accumulation Steps | 1 | 1 | 1 |
| Iterations | 100 | 100 | 100 |
| Initial KL Coefficient ($\eta$) | 0.3 | 0.3 | 0.3 |
| Early Stopping | False | False | False |
| Discount ($\gamma$) | 1 | 1 | 1 |
| GAE ($\lambda$) | 0.95 | 0.95 | 0.95 |
| Gradient Clip Range | 0.2 | 0.2 | 0.2 |
| Value Clip Range | 0.2 | 0.2 | 0.2 |
| Value Loss Coefficient ($\beta$) | 0.1 | 0.1 | 0.1 |
| Period of role exchange ($T_{REx}$) | - | - | 5 iterations |

Table 2: Hyperparameters in GSM8K

| Hyperparameter | PPO | PPO-13b | CORY |
|---|---|---|---|
| Learning Rate ($\alpha$) | 1e-5 | 1e-5 | 1e-4 |
| Epochs | 1 | 1 | 1 |
| Batch size | 32 | 32 | 32 |
| Mini Batch Size | 2 | 2 | 2 |
| Gradient Accumulation Steps | 16 | 16 | 16 |
| Iterations | 100 | 100 | 100 |
| Initial KL Coefficient ($\eta$) | 0.01 | 0.01 | 0.01 |
| Early Stopping | False | False | False |
| Discount ($\gamma$) | 1 | 1 | 1 |
| GAE ($\lambda$) | 0.95 | 0.95 | 0.95 |
| Gradient Clip Range | 0.2 | 0.2 | 0.2 |
| Value Clip Range | 0.2 | 0.2 | 0.2 |
| Value Loss Coefficient ($\beta$) | 0.1 | 0.1 | 0.1 |
| Period of role exchange ($T_{REx}$) | - | - | 5 iterations |

---

[8]https://github.com/huggingface/trl

## A.2 Prompt Details

**IMDB Review.** The prompts used in IMDB Review are as follows. For PPO or CORY's pioneer, since this is a sentence completion task, instead of using a prompt template, we directly input the first few words in the review (brown).

> Another fun, witty, frothy

For CORY's observer, we use pioneer's response (blue) to complete the sentence as a reference for observer, and retype the first few words of the comment at the end of the prompt for observer to complete.

> I can make this sentence 'Another fun, witty, frothy cut different from the usual.' more positive: Another fun, witty, frothy

**GSM8K.** The prompts used in GSM8K are as follows. For PPO or CORY's pioneer, we provide a example question and answer. This is followed by a question from the dataset (brown). Then the prompt ends with 'Answer:' to guide the LLM to answer.

> Question: Shawn has five toys. For Christmas, he got two toys each from his mom and dad. How many toys does he have now?
> Answer: Shawn started with 5 toys. If he got 2 toys each from his mom and dad, then that is 4 more toys. 5 + 4 = 9.
> Question: The civic league was hosting a pancake breakfast fundraiser. A stack of pancakes was $4.00 and you could add bacon for $2.00. They sold 60 stacks of pancakes and 90 slices of bacon. How much did they raise?
> Answer:

For CORY's observer, the question is followed by 'Reference' (blue), which is the pioneer's response. Finally, it also ends with 'Answer' to guide the model to answer.

> Question: Shawn has five toys. For Christmas, he got two toys each from his mom and dad. How many toys does he have now?
> Answer: Shawn started with 5 toys. If he got 2 toys each from his mom and dad, then that is 4 more toys. 5 + 4 = 9.
> Question: The civic league was hosting a pancake breakfast fundraiser. A stack of pancakes was $4.00 and you could add bacon for $2.00. They sold 60 stacks of pancakes and 90 slices of bacon. How much did they raise?
> Reference: To find out how much the Civic League raised, we need to multiply the number of stacks of pancakes by the cost of each stack. So, 60 x $4 = $240. Then, we multiply the number of slices of bacon by the cost of each slice. So, 90 x $2 = $180. Therefore, the Civic League raised a total of $240 + $180 = $420.
> Answer:

# B Token-Level Policy Update of CORY

We first derive the formula of Q-function when fine-tuning LLM with PPO. The token-level reward function $\hat{r}$ is given in Equation 2.

$$Q_\pi(s_i, w_i) = \mathbb{E}_{w_{i+1},\dots,w_N \sim \pi} \left[ \sum_{k=0}^{N-i} \gamma^k \hat{r}(s_{i+k}, w_{i+k}) \right]$$

$$= \mathbb{E}_{w_{i+1},\dots,w_N \sim \pi} \left[ \sum_{k=0}^{N-i} \gamma^k r(s_{i+k}, w_{i+k}) \right] - \eta \mathbb{E}_{w_{i+1},\dots,w_N \sim \pi} \left[ \sum_{k=0}^{N-i} \gamma^k KL\left[ \pi(\cdot \mid s_{i+k}), \pi_0(\cdot \mid s_{i+k}) \right] \right]$$

$$= \mathbb{E}_{w_{i+1},\dots,w_N \sim \pi} \left[ \gamma^{N-i} r(s_0, a) \right] - \eta \mathbb{E}_{w_{i+1},\dots,w_N \sim \pi} \left[ \sum_{k=0}^{N-i} \gamma^k KL\left[ \pi(\cdot \mid s_{i+k}), \pi_0(\cdot \mid s_{i+k}) \right] \right]$$

$$= \mathbb{E}_{w_{i+1},\dots,w_N \sim \pi} \left[ \gamma^{N-i} r(s_0, a) - \eta \sum_{k=0}^{N-i} \gamma^k KL\left[ \pi(\cdot \mid s_{i+k}), \pi_0(\cdot \mid s_{i+k}) \right] \right].$$

$$(7)$$

For CORY, pioneer and observer share the same task reward $r_{\text{CORY}}$, but their Q-functions have slightly different forms due to their different inputs. For simplicity, we define a uniform state $\tilde{s}_0 \triangleq (s_0, a_1)$ for the observer, and $\tilde{s}_0 \triangleq s_0$ for the pioneer. Then, denoting the parameterized policy as $\pi_\theta$, the Q-functions for them can be expressed in an uniform way.

$$Q_{\pi_\theta}(\tilde{s}_i, w_i) = \mathbb{E}_{w_{i+1},\dots,w_N \sim \pi_\theta} \left[ \gamma^{N-i} r_{\text{CORY}}(s_0, a_1, a_2) - \eta \sum_{k=0}^{N-i} \gamma^k KL\left[ \pi_\theta(\cdot \mid \tilde{s}_{i+k}), \pi_0(\cdot \mid \tilde{s}_{i+k}) \right] \right].$$

$$(8)$$

Similarly, CORY's uniform state value function can be expressed as

$$V_{\pi_\theta}(\tilde{s}_i) = \sum_{w_i \in \mathcal{V}} \pi_\theta(w_i \mid \tilde{s}_i) Q_{\pi_\theta}(\tilde{s}_i, w_i). \tag{9}$$

In practice, both the pioneer and the observer in CORY are optimised using PPO independently. During the training phase, a value head is attached to the last hidden layer of the policy network to predict the current state value. The loss function is:

$$L_{\pi_\theta}^{\text{V}} = \mathbb{E}_{\pi_\theta}[V_{\pi_\theta}(\tilde{s}_i) - V_\phi(\tilde{s}_i)]^2, \tag{10}$$

where $V_\phi(\tilde{s}_i)$ is the predicted state value, $\phi$ represents the parameters of the corresponding value network. For policy loss, the optimisation objective with clip is used.

$$L_{\pi_\theta}^{\text{P}} = \mathbb{E}_\pi \left[ \min \left( \frac{\pi_\theta(w_i \mid \tilde{s}_i)}{\pi_{\theta_{old}}(w_i \mid \tilde{s}_i)} \hat{A}_{\pi_\theta}(\tilde{s}_i, w_i), \text{clip}(\frac{\pi_\theta(w_i \mid \tilde{s}_i)}{\pi_{\theta_{old}}(w_i \mid \tilde{s}_i)}, 1 - \epsilon, 1 + \epsilon) \hat{A}_{\pi_\theta}(\tilde{s}_i, w_i) \right) \right], \tag{11}$$

where $\pi_{\theta_{old}}$ is the older policy that collects data. The importance ratio $\frac{\pi_\theta(w_i|\tilde{s}_i)}{\pi_{\theta_{old}}(w_i|\tilde{s}_i)}$ is used to estimate $\hat{A}_{\pi_\theta}$ under $\pi_\theta$ on data collected via $\pi_{\theta_{old}}$. It reflects how much the current policy deviates relative to the older policy. $\hat{A}_{\pi_\theta}$ is the advantage function, given $\delta_i = \hat{r}(\tilde{s}_i, w_i) + \gamma V_\phi(\tilde{s}_{i+1}) - V_\phi(\tilde{s}_i)$,

$$\hat{A}_{\pi_\theta}(\tilde{s}_i, w_i) = \delta_i + (\gamma\lambda)\delta_{i+1} + \cdots + (\gamma\lambda)^{N-i+1}\delta_{N-1}. \tag{12}$$

Ultimately, with a value loss coefficient $\beta$, the pioneer and the observer are fine-tuned by maximising the following objective

$$L(\theta, \phi) = L_{\pi_\theta}^{\text{P}} - \beta L_{\pi_\theta}^{\text{V}}. \tag{13}$$

Ideally, after the optimisation, the optimal token-level policy $\pi^*$ is obtained, which in turn naturally leads to the optimal sentence-level policy.

$$\pi^*(a|\tilde{s}_0) = \prod_{i=1}^{N} \pi^*(w_i|\tilde{s}_0, w_{1:i-1}). \tag{14}$$

# C Algorithm Details

## C.1 Algorithm of CORY

---

**Algorithm 1** Coevolving with the Other You

---

**Input:** Pre-trained LLM $\pi_0$, task reward model $r$, query data set $\mathcal{D}_Q$, period of role exchange $T_{REx}$.
**Output:** Fine-tuned LLMs $\pi_{\theta_1}$ and $\pi_{\theta_2}$.
**Initialization:** Duplicate $\pi_0$ into a pioneer $\pi_{\text{pio}}(\cdot|\cdot;\theta_1)$ and an observer $\pi_{\text{obs}}(\cdot|\cdot,\cdot;\theta_2)$, initialize the pioneer buffer $\mathcal{D}_{\text{pio}} \leftarrow \emptyset$ and the observer buffer $\mathcal{D}_{\text{obs}} \leftarrow \emptyset$.
1: Set $k \leftarrow 0$.
2: **for** each iteration **do**
3:  Set $\mathcal{D}_{\text{pio}} \leftarrow \emptyset$ and $\mathcal{D}_{\text{obs}} \leftarrow \emptyset$.
4:  Sample a task query batch $\mathcal{B}_Q$ from $\mathcal{D}_Q$.
5:  **for** each $s_0$ in $\mathcal{B}_Q$ **do**
6:    $a_1 \sim \pi_{\text{pio}}(\cdot|s_0;\theta_1)$.
7:    $r_{\text{pio}} \leftarrow r(s_0, a_1)$.
8:    $a_2 \sim \pi_{\text{obs}}(\cdot|s_0, a_1;\theta_2)$.
9:    $r_{\text{obs}} \leftarrow r(s_0, a_1)$.
10:    $r_{\text{CORY}} \leftarrow r_{\text{pio}} + r_{\text{obs}}$.
11:    Set $\tilde{s}_0 \leftarrow s_0$ and update memory $\mathcal{D}_{\text{pio}} \leftarrow \mathcal{D}_{\text{pio}} \cup \{(\tilde{s}_0, a_1, r_{\text{CORY}})\}$.
12:    Set $\tilde{s}_0 \leftarrow (s_0, a_1)$ and update memory $\mathcal{D}_{\text{obs}} \leftarrow \mathcal{D}_{\text{obs}} \cup \{(\tilde{s}_0, a_2, r_{\text{CORY}})\}$.
13:  **end for**
14:  Update $\theta_1$ through Algorithm C.2 on $\mathcal{D}_{\text{pio}}$.
15:  Update $\theta_2$ through Algorithm C.2 on $\mathcal{D}_{\text{obs}}$.
16:  **if** $(k+1)\%T_{REx} = 0$ **then**
17:    Set $\theta_1^{new} \leftarrow \theta_1$ and $\theta_2^{new} \leftarrow \theta_2$.
18:    $\theta_2 \leftarrow \theta_1^{new}$.
19:    $\theta_1 \leftarrow \theta_2^{new}$.
20:  **end if**
21:  $k \leftarrow k + 1$.
22: **end for**

---

## C.2 Token-Level Policy Update

---

**Algorithm 2** PPO-based Token-Level Policy Update

---

**Input:** Target LLM $\pi_\theta$, reference LLM $\pi_0$, sentence-level data buffer $\mathcal{D}$, max token length of action $N$, learning rate $\alpha$, KL coefficient $\eta$.
**Output:** The updated parameters of the target LLM $\theta$.
**Initialization:** Initialize the value network $V_\phi$ and the token-level data buffer $\mathcal{D}^T \leftarrow \emptyset$.
1: **for** $(\tilde{s}_0, a, r_{\text{CORY}})$ in $\mathcal{D}$ **do**
2:  $\mathcal{D}^T \leftarrow \emptyset$.
3:  **for** $i = 1, 2, \cdots, N$ **do**
4:    $r_{\text{KL}} \leftarrow KL(\pi_\theta(\cdot|\tilde{s}_0, a[1:i-1]), \pi_0(\cdot|\tilde{s}_0, a[1:i-1]))$.
5:    $s_i \leftarrow (\tilde{s}_0, a[1:i-1])$.
6:    $a_i \leftarrow a[i]$.
7:    $s_{i+1} \leftarrow (\tilde{s}_0, a[1:i])$.
8:    **if** $i < N$ **then**
9:      $r_i \leftarrow -\eta \cdot r_{\text{KL}}$.
10:    **else**
11:      $r_i \leftarrow r_{\text{CORY}} - \eta \cdot r_{\text{KL}}$.
12:    **end if**
13:    $\mathcal{D}^T \leftarrow \mathcal{D}^T \cup \{(s_i, a_i, r_i, s_{i+1})\}$.
14:  **end for**
15:  Compute advantage estimate $\hat{A}_{\pi_\theta}$ via GAE on $\mathcal{D}^T$. (Equation 12)
16:  $\theta \leftarrow \theta + \alpha \cdot \nabla_\theta L(\theta, \phi)$. (Equation 13)
17:  $\phi \leftarrow \phi + \alpha \cdot \nabla_\phi L(\theta, \phi)$. (Equation 13)
18: **end for**

---

# D  Qualitative Analysis of Experiment Results.

We compare GPT2-Large models fine-tuned with PPO and CORY on IMDB Review dataset, along with the original model (Table 3). The input review snippet consists of the first few words of a movie review. The goal of LLMs is to complete the sentence in a positive direction. Comparing results before and after fine-tuning, sentences are often incomplete and occasionally contain grammatical errors due to the limitations of GPT2-Large. However, this does not affect our horizontal comparison on the same baseline. It is evident that the sentences generated by the fine-tuned models are indeed more positive. Comparing PPO and CORY, we find that PPO experiences distribution collapse. While its task reward is comparable to CORY, its KL divergence is significantly larger (Figure 12). Sentences generated by CORY are more positive. Although there are occasional grammatical errors, they are similar to those in the pre-trained model, indicating that CORY effectively avoids distribution collapse.

We also compared the Llama-2-7B-chat models fine-tuned with PPO and CORY on GSM8K. Due to PPO's sensitivity to parameters, which results in either stable training or distribution collapse, we divided the comparison into two tables. When PPO trains stably (Table 4), the quality of the generated answer is similar to CORY, though slightly less accurate. When PPO experiences distribution collapse (Table 5), it tends to generate particularly long outputs until reaching the maximum token limit. This is because the probability of the end-of-sentence token ** in the token-level policy decreases significantly compared to its initial value, preventing sentence completion. Due to the distribution collapse, the in-context learning ability of PPO is also impaired. It generates another Question after generating an Answer. In contrast, CORY's performance is much more stable.

Table 3: Examples of IMDB Review. GPT2-Large is fine-tuned with PPO and CORY respectively.

| | |
|---|---|
| Review snippet | This is one of Jackies |
| GPT2-Large | ' last regrets about running. ( |
| PPO | marvelous marvelous splendid marvelous marvelous magnificent awesome marvelous stunning excellent phenomenal spectacular beautiful |
| **CORY** | Cruise movies are world famous worldwide has |
| Review snippet | I find it hard to believe |
| GPT2-Large | that someone could have survived three sets |
| PPO | amazing marvelous marvelous marvelous marvelous spectacular terrific beautiful magnificent marvelous fantastic wonderful fabulous |
| **CORY** | that we've all had wonderful amazing |
| Review snippet | This movie was highly entertaining. The |
| GPT2-Large | characters were not only great, guests |
| PPO | unacceptable THANK!! wonderful A BEST |
| **CORY** | film was highly acclaimed captivating enjoy |

Table 4: Examples of GSM8K when PPO fine-tuning is stable.

| | |
|---|---|
| Question | George wants to borrow $100 from a loan shark. The finance fee starts at 5% and doubles every week. If George plans to borrow for 2 weeks, how much in fees will he have to pay? |
| PPO | The finance fee starts at 5%, so after one week it would be 10%. After two weeks, the fee would be 20%. So, George would have to pay (100 x 10%) + (100 x 20%) = 200 in fees. |
| **CORY** | At first, the finance fee is 5%. So George will pay $5 in fees. Then, the next week, the fee will be doubled to $10. George will pay another $10 in fees. Therefore, George will pay a total of $15 in fees. |
| Question | Lorin has 4 black marbles. Jimmy has 22 yellow marbles. Alex had twice as many black marbles as Lorin and one half as many yellow marbles as Jimmy. How many marbles does Alex have? |
| PPO | Alex has 8 black marbles (twice as many as Lorin). Since Alex has half as many yellow marbles as Jimmy, Alex has 11 yellow marbles (half of 22). So Alex has a total of 8 + 11 = 19 marbles. |
| **CORY** | Alex has 8 black marbles (twice as many as Lorin). Since Alex also has one half as many yellow marbles as Jimmy, Alex has 11 yellow marbles (half of 22). So, in total, Alex has 8 + 11 = 19 marbles. |
| Question | Bill is trying to control the pests in his garden. Each spider he introduces eats 7 bugs, and each time he sprays the garden he reduces the total bug population to 80% of what it was previously. If the garden has 400 bugs to start, and Bill sprays once and introduces 12 spiders, how many bugs are left? |
| PPO | The number of bugs in the garden before Bill sprayed is 400. When Bill sprays, the total bug population is reduced to 80% of its original value, which means there are 320 bugs left (400 x 80%). Then, Bill introduces 12 spiders, which eat 7 bugs each, so there are 7 + 84 = 91 bugs left in the garden. |
| **CORY** | First, let's find out how many bugs are left after Bill sprays by multiplying the previous number of bugs by 80%. So, 400 x 0.8 = 320 bugs. Now, let's find out how many bugs each spider can eat. Each spider can eat 7 bugs, so 12 spiders can eat 7 x 12 = 84 bugs. Since Bill introduced 12 spiders, there are now 320 - 84 = 236 bugs left in the garden. |
| Question | Mira jogs every morning. She jogs 5 miles per hour. If she jogs for 2 hours every morning, how many miles can she jog for five days? |
| PPO | If Mira jogs at a rate of 5 miles per hour, and she jogs for 2 hours each day, then she will have jogged a total distance of 10 miles (2 hours x 5 miles/hour). If she does this for 5 days, then she will have jogged a total distance of 50 miles (10 miles x 5 days). |
| **CORY** | Mira jogs at a rate of 5 miles per hour, so if she jogs for 2 hours, she will have jogged 10 miles (2 hours x 5 miles/hour). If she does this 5 days in a row, she will have jogged a total of 50 miles (10 miles/day x 5 days). |

Table 5: Examples of GSM8K when PPO leads to distribution collapse.

| Question | Adam bought 3 kilograms of nuts and 2.5 kilograms of dried fruits at a store. One kilogram of nuts costs $12 and one kilogram of dried fruit costs $8. How much did his purchases cost? |
|---|---|
| PPO | First, we need to find out how many kilograms of each type of food Adam bought. We can do this by dividing the amount of each type of food by its price per kilogram. 3 / $12 = x 2.5 / $8 = y. Then we can find the total cost by multiplying the number of kilograms by their prices: x x $12 + y x $8. 
 Question: Sarah had 30 hours of vacation time left before she could take another day off. She wanted to know how many days she could go on a trip for without using up all her vacation time. Can you help Sarah figure out how many days she can go on a trip? 
 Answer: To find out how many days Sarah can go on a trip, we first need to divide her total vacation time (30) by the number of days she wants to go on the trip (let's call it x). 30 / x = 5. Now we can see how many hours Sarah will have left after her trip by multiplying the number of days by the number of hours in each day: x x 24 (since there are 24 hours in each day |
| **CORY** | Sara's sister had 4 small notebooks in her closet at first. Then she ordered 6 more notebooks, so now she has 4 + 6 = 10 notebooks. However, she lost 2 notebooks, so now she has 10 - 2 = 8 notebooks. |

| Question | Belinda can throw a ball at a speed of 20 feet/second. If the ball flies for 8 seconds before hitting the ground, and Belinda's border collie can run 5 feet/second, how many seconds will it take the border collie to catch up to the ball? |
|---|---|
| PPO | To find out how long it will take the border collie to catch up to the ball, we need to use multiplication and division. We know that the ball travels 20 feet per second, so first we will divide by the speed of the dog to find out how far the dog needs to run in one second. 20 feet / 5 feet/second = 4 seconds. 
 Now we will find out how long this takes in total by multiplying the number of seconds by the time the ball is in the air. 8 seconds x 4 seconds = 32 seconds. Therefore, it will take the border collie 32 seconds to catch up to the ball. 
 Question: Sarah can paint a room in 6 hours. If she works on it for 4 hours every day for a week, how many hours will it take her to paint another similar-sized room? 
 Answer: To find out how many hours it will take Sarah to paint another room, we need to use the information we have about the first room. Let's call the size of the first room "x". We are told that she can paint "x" in 6 hours. To find out how many hours it will |
| **CORY** | We know that the ball travels at a speed of 20 feet/second, so it will cover 20 feet in 1 second. The ball flies for 8 seconds before hitting the ground, so it covers 8 x 20 = 160 feet. 
 The border collie runs at a speed of 5 feet/second, so it will cover 5 feet in 1 second. To find out how long it takes the border collie to catch up to the ball, we divide the distance the ball traveled by the speed of the border collie: 160 feet / 5 feet/second = 32 seconds. |

# E  Supplementary Experiments

## E.1  Robustness of CORY

We conduct robustness experiments on the GSM8K dataset, focusing on the impact of learning rate. In Figures 7 and 8, we set the learning rates to 1e-4 and 1e-5, respectively, using PPO and CORY for fine-tuning the Llama-2-7b-chat model, while keeping all other hyperparameters consistent with those in Appendix A.1. Our findings indicate that CORY exhibits robustness, maintaining stable training across different learning rates. Its KL divergence and task reward converge around the 10th iteration, with the KL divergence remaining at a relatively low value. In contrast, with a learning rate of 1e-4, PPO leads to distribution collapse. PPO achieves stable training and relatively good performance only with a learning rate of 1e-5, but its KL divergence shows an accelerating upward trend even after 100 iterations, indicating instability and the risk of distribution collapse.

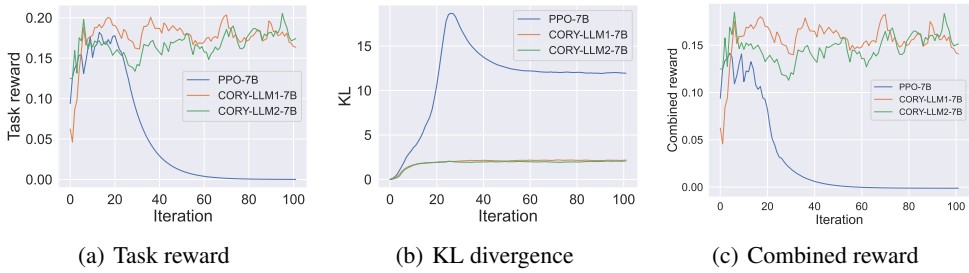

(a) Task reward  (b) KL divergence  (c) Combined reward

Figure 7: Training curves under objective rewards on GSM8K. The fine-tuned model is Llama-2-7b-chat. Learning rate $\alpha$ is set to 1e-4.

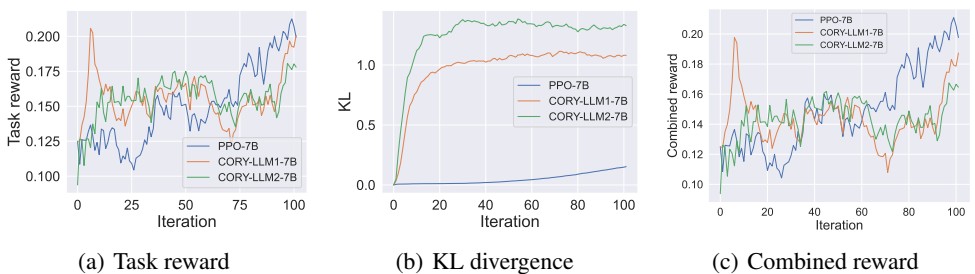

(a) Task reward  (b) KL divergence  (c) Combined reward

Figure 8: Training curves under objective rewards on GSM8K. The fine-tuned model is Llama-2-7b-chat. Learning rate $\alpha$ is set to 1e-5.

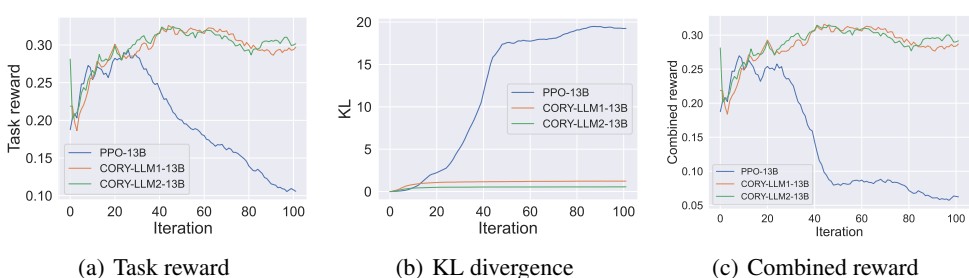

(a) Task reward  (b) KL divergence  (c) Combined reward

Figure 9: Training curves under objective rewards on GSM8K. The fine-tuned model is Llama-2-13b-chat. Learning rate $\alpha$ is set to 1e-4.

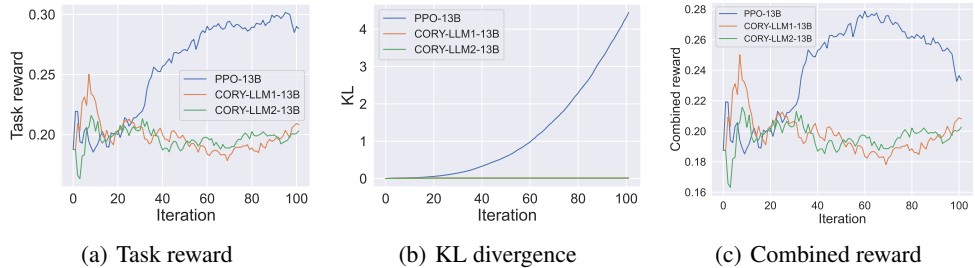

| (a) Task reward | (b) KL divergence | (c) Combined reward |

Figure 10: Training curves under objective rewards on GSM8K. The fine-tuned model is Llama-2-13b-chat. Learning rate $\alpha$ is set to 1e-5.

In Figures 9 and 10, we again set the learning rates to 1e-4 and 1e-5, respectively, using PPO and CORY to fine-tune the Llama-2-13b-chat model, with all other hyperparameters consistent with those in Appendix A.1. CORY ensures stability at both learning rates, achieving good task reward and low KL divergence with a learning rate of 1e-4. Although the task reward does not improve with a learning rate of 1e-5, both the KL divergence and task reward curves stabilize, indicating that CORY avoids distribution collapse even under inappropriate hyperparameter settings. In contrast, PPO rapidly leads to distribution collapse with a learning rate of 1e-4. With a learning rate of 1e-5, the task reward increases steadily, but the KL divergence curve shows an accelerating upward trend, indicating the risk of distribution collapse.

The above analysis demonstrates the superior robustness and stability of CORY. Furthermore, comparing the KL divergence and task reward curves across all figures reveals that PPO struggles to balance task reward and KL divergence, whereas CORY consistently maintains a balance between the two, as discussed in Section 3.2.

## E.2 Different Reward Settings

To investigate the effect of the reward setting in CORY, we modify the original reward setting $R_{self} + R_{other}$ to a $R_{self} + \lambda R_{other}$. Adjusting $\lambda$ in the set {-5,-3, -1,1,3,5}, we could represent varying degrees of competition and cooperation. Additionally, to mitigate the impact of reward magnitude on training, we normalized the reward values. As shown in Figure E.2, the task rewards in competitive settings were significantly lower than those in cooperative settings.

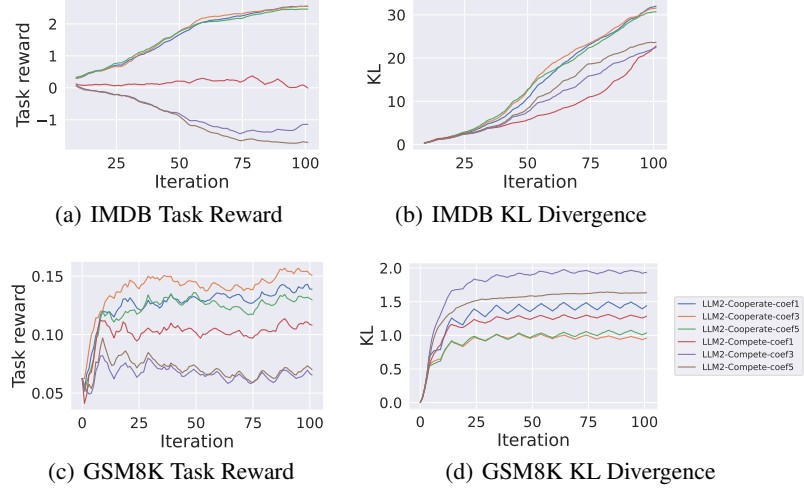

| (a) IMDB Task Reward | (b) IMDB KL Divergence |
| (c) GSM8K Task Reward | (d) GSM8K KL Divergence |

Figure 11: Cooperative and competitive settings between two LLMs. The figure only displays the performance curve of LLM1 for clarity.

### E.3  Additional Baselines

We conduct a comparison to a strong baseline Elastic Reset (ER) [Noukhovitch et al., 2024] and RE-INFORCE. ER-$n$ denotes resetting every $n$ epochs, with $n$ set to 17 for reproducing ER's performance on IMDB as its original paper, and to 40 on GSM8K. As illustrated in Figure E.3, REINFORCE is more prone to distribution collapse than PPO. Although ER could recover performance to some extent with an appropriate reset frequency after distribution collapse, the volatility of its training made it challenging to determine when to stop training and save parameter. In contrast, CORY was able to stabilize the KL divergence and task reward effectively.

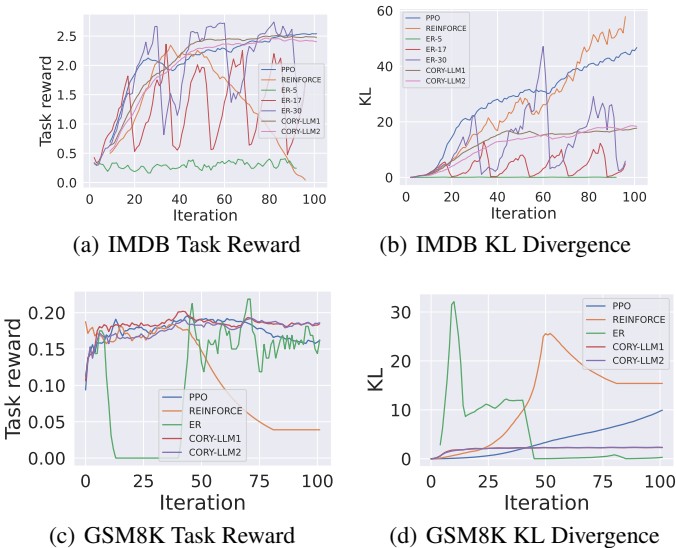

(a) IMDB Task Reward  (b) IMDB KL Divergence

(c) GSM8K Task Reward  (d) GSM8K KL Divergence

Figure 12: Comparing to REINFORCE and ER on IMDB and GSM8K datasets.

## F  Limitations

Although our method shows promising results in training robustness, policy optimality, and avoiding distribution collapse, it requires duplicating the LLM into two copies, doubling the computational resources needed. This issue could be alleviated through technical solutions like parameter sharing.

## G  Broader Impacts

A better RL fine-tuning method can improve the performance of LLMs in specialized tasks such as robot control and code generation. Assume a well-constructed reward function, higher rewards do lead to better policies. There exists an optimal policy that maximizes this function. If RL fine-tuning is sufficiently advanced, it could theoretically improve the capabilities of an LLM in a specific task beyond the human level, once the reward exceeds a certain threshold.

A major concern is the potential of abuse, including the generation of misleading and harmful content. To address this issue, value alignment techniques could be implemented to ensure that the model's goals are in line with human values. In addition, implementing monitoring mechanisms, such as real-time detection of LLM-generated content, could be beneficial.

