# OpenReview forum: "Coevolving with the Other You: Fine-Tuning LLM with Sequential Cooperative Multi-Agent Reinforcement Learning"
_NeurIPS.cc/2024/Conference — NeurIPS 2024 poster_

### Official Review · Reviewer_chZh · 2024-06-27

**Soundness:** 2
**Presentation:** 3
**Contribution:** 3
**Rating:** 6
**Confidence:** 4

**Summary:**

The paper introduces CORY, a novel reinforcement learning (RL) technique for fine-tuning language models that casts a multi-agent framework on the trained language model by duplicating it at initialization and then using the two copies to improve one another.
Essentially, a "pioneer" gives the first guess, used as a reference to an "observer" which gives the final answer, then both agents are rewarded for both of their outputs in a cooperative fashion and trained independently with an underlying RL algorithm, here PPO.
The authors claim that CORY improves on single-agent PPO in terms of downstream performance, training stability, and robustness to distribution collapse.

**Strengths:**

The paper is easy to follow and includes enough details about the experiments.
Casting the problem into a multi-agent framework to change the optimization dynamics is original and will likely influence other work to build on it.
The observations made by the authors about the multi-agent framework changing the dynamics training dynamics to make the observer more easily optimize for the reward while maintaining a low KL are interesting.
The method can potentially be applied on top of any RL algorithm, either with independent learners as shown in the paper or by introducing centralized training critic network sharing, etc.

**Weaknesses:**

Overall, I find the contribution of the paper to be original and likely to have impact, however, I also find a number major issues in the work that make it not ready for publication at this venue.

Unclear contextualization and missing baselines:
- One main contribution of the paper is that it finds a policy that has a better Pareto frontier while still using the sum of task reward and KL divergence, but the KL divergence is an artificial reward specific to current RL fintuning methods in language modeling. If the paper's method is to be highlighted from a language modeling perspective then I would expect it to compare to methods like [1] which also claim to obtain a better Pareto frontier. And more than that, I would expect the method to show its benefits in downstream tasks for example with AlpacaFarm [2] as although a better frontier could be seen a a proxy to robustness to model collapse, it isn't clear if it translates to better language modeling capabilities.
Otherwise if the method is to be highlighted in general multi-objective RL fine-tuning tasks, then I would expect comparison to multi-objective RL methods.
- (minor) The method can also be seen a self-improvement method, so the pepper would benefit from discussing and contrasting methods from the same family.

An insufficient number of benchmarks:
- The authors claim to evaluate their method "across diverse reward function landscapes, encompassing both subjective and objective categories", but only one dataset per category is tested. Judging by the scope of the claims the authors make and that this is the only form of evidence provided, I would expect at least 3 or 5 datasets per category.

Hyperparameter choice:
- The authors use a single set of hyperprameters, with no apparent justification. This is not enough to make claims about a method being better than another.
- (minor) PPO with 1 epoch and 1 batch per epoch (as in the hyperprameter table) becomes just a policy gradient, so the authors seem to be effectively using a vanilla actor-critic with a GAE value estimator and (usually useless) value clipping.

Presentation:
- (minor) The wording of the claims made in the abstract and introduction "policy optimality, "resilience to distribution collapse", and "robustness during training" is not reused in the experiments section which makes it hard to connect evidence to claims.
- (minor) The $\geq$ symbol in equation 6 between vector-values objectives is undefined. One would expect a definition of Pareto dominance.

[1] Noukhovitch, Michael, et al. "Language model alignment with elastic reset." Advances in Neural Information Processing Systems 36 (2024).
[2] Dubois, Yann, et al. "Alpacafarm: A simulation framework for methods that learn from human feedback." Advances in Neural Information Processing Systems 36 (2024).

**Questions:**

How did the authors select the hyperprameters for their methods?

**Limitations:**

Although in the appendix, the authors adequately state the limitations of their work and its broader impacts.

---

> ### Author Rebuttal · Authors · 2024-08-07
>
> Thank you for your valuable feedback and time spent! We’re glad you found our idea original and likely to influence other work to build on it, our paper easy to follow, and the observations from the multi-objective RL perspective interesting. We would like to address your concerns below.
>
> ## **R4-1** Missing Baselines
> We appreciate your concern regarding the need for more convincing baselines. In response, we have added an additional baseline, Elastic Reset(ER). The original ER paper also tested using the IMDB dataset. We adopted the experimental setup from the paper with parameters set to decay=0.995, reset_freq=17, and eta=0.001, successfully replicating their results (ER-17). Moreover, we conducted comparisons with two other sets of reset frequencies. The specific experimental results are as follows:
>
> |     | Task-reward ↑ | KL divergence ↓ |
> |-----|---------------|-----------------|
> | PPO | 2.17          | 44.33           |
> | ER-30 | 2.65       | 32.15          |
> | ER-17 | 2.63       | 21.73           |
> | ER-5 | 0.53       | 0.32          |
> | CORY | **2.67**    | **15.18**      |
>
> Furthermore, due to the reset mechanism, the training curve of Elastic Reset (ER) exhibits a peak-like shape, unlike the stable state observed in our case.
>
> ## **R4-2** Insufficient Benchmarks
> Thank you for the reminder. We will amend the statement "across diverse reward function landscapes, encompassing both subjective and objective categories" in the revised manuscript.
>
> Regarding the benchmarks, we believe they are currently persuasive enough. Reviewer Syvc mentioned that "the experiments are adequate and effectively support the proposed method," and Reviewer F4Ro also noted that we have utilized modern and relevant benchmarks and models.
>
> However, we understand your concerns regarding the benchmarks. Therefore, we have included an additional, widely used dataset, Anthropic-HH. On this dataset, CORY shows its superior ability to balance task reward and KL divergence than all the baselines (PPO, Elastic reset, REINFORCE). We hope this can addresses your concerns. Training curves are in Figure 4 in the additional PDF.
>
> ## **R4-3** Self-improvement Method
> In the domain of LLM fine-tuning, there exist some self-improvement methods, such as SPIN [1] and Iterative DPO [2]. However, these approaches are offline algorithms in supervised manner, where self-improvement stems from making more comprehensive use of the dataset.
>
> In contrast, RL as an online algorithm, is optimized on data generated by itself. In the MARL field, the success of AlphaGO and OpenAI Five has validated the effectiveness of the self-improvement among a multi-agent system that diverse data are generated through competition/cooperation among multiple agents. The problem we are concerned with is how to make the RL fine-tuning of LLM benefit from this self-improvement in multi-agent system [3, 4]. Therefore, CORY introduces a completely different self-improvement mechanism into the fine-tuning of LLMs.
>
> [1] Chen, Zixiang, et al. "Self-Play Fine-Tuning Converts Weak Language Models to Strong Language Models." Forty-first International Conference on Machine Learning.
>
> [2] Yuan, Weizhe, et al. "Self-Rewarding Language Models." Forty-first International Conference on Machine Learning.
>
> [3] A social network for AI. Nat Mach Intell 5, 1175 (2023). https://doi.org/10.1038/s42256-023-00769-4
>
> [4] Duéñez-Guzmán, Edgar A., et al. "A social path to human-like artificial intelligence." Nature Machine Intelligence 5.11 (2023): 1181-1188.
> ## **R4-4** Hyperparameter Choice
> The parameter settings for fine-tuning GPT2 followed the default configuration in TRL for the IMDB dataset, while the fine-tuning for LlaMA2 primarily adhered to the guidelines established by StackLlama. In pursuit of a fair comparison, all parameters were carefully chosen to balance the stability and performance of PPO. And a grid search was conducted over learning rates and eta coefficients within the ranges lr[1e-6, 1e-5, 1e-4] and eta[1e-3, 1e-2, 1e-1, 0.2, 0.3] respectively, to select the parameters that yielded the most stable training for PPO. Given CORY's robustness to parameter variations (as evidenced by the results of adjusting learning rates and kl coefficients presented in Appendix E), the parameters for PPO, with the exception of the learning rate, were directly applied to CORY. In the GSM8K dataset, we adjusted the learning rate for CORY. Important parameters include lr and kl_coef, while parameters such as mini_batch, ppo_epoch are considered to be of relative lesser importance.
>
> ## **R4-5** PPO Hyperparameter Concern
> Yes, it is acknowledged that a strictly on-policy PPO is equivalent to A2C. However, the hyperparameter 'Epoch' in our hyperparameter table does not refer to PPO epochs. In our experiments, the ppo_epochs were actually set to 4, which aligns with the standard configuration of PPO. We will update our parameter table to prevent similar misunderstandings.
> ## **R4-6** Feedback on the Presentation
> Thanks for your suggestion! Regarding your first point, our paper already contains clear experiments related to "policy optimality" (Sections 4.1 & 4.2), "resilience to distribution collapse" (Sections 4.1 & 4.2), and "robustness during training" (Appendix E). We will explicitly add these descriptions to the experimental section of our paper.
>
> As for your second point, we will include definitions of the ">=" symbol in the revised manuscript.

---

> > ### Comment · Reviewer_chZh · 2024-08-11
> >
> > I thank the authors for the clarifications. Most of my concerns have been addressed.
> > I am increasing my score to 5 as I still believe the submission would benefit from better contextualization (should it be compared to multi-objective methods?) and more baselines (I appreciate the authors comparing to ER).

---

> > > ### Author Response · Authors · 2024-08-13
> > >
> > > Thank you for your timely response! We're glad to have addressed most of your concerns. We'd like to provide further clarification on multi-objective RL and baselines.
> > >
> > > ### 1. **Clarification on multi-objective reinforcement learning (MORL)** ###
> > >
> > > From the perspective of RL, the RL fine-tuning of LLM indeed constitutes a multi-objective (specifically, dual-objective) RL problem. Algorithms specifically designed for MORL can be categorized into two main types: multi-policy methods and single-policy methods [1]. Multi-policy methods train multiple policies, each corresponding to a single combination of objectives [2]. This approach equates to altering the weights of KL and task reward in Section 3.2(constructing multiple combinations of objectives), training multiple policies through conventional RL methods, and selecting the more effective policies. However, as demonstrated in Fig.2, the performance of all policies (their Pareto frontier) is inferior to policy trained under any weight by CORY.
> > >
> > > On the other hand, single-policy methods address multi-objective problems through reward scalarization [2]. In our paper, the baseline such as PPO, considers the weighted loss of both the task reward and KL during training, even dynamically adjusting the weight [3], which essentially represents a naive form of reward scalarization. To further alleviate your concerns, we conducted experiments on the IMDB dataset using GGF-PPO [4], which is one of the SOTA algorithms in the MORL domain. As demonstrated in the attached Table 1, GGF-PPO shows only a marginal performance improvement over PPO (a 3.2% increase in Reward and a 13.3% reduction in KL). Therefore, the effectiveness of single-policy methods is also limited. Compared to CORY, they fundamentally cannot leverage the interaction and emergent intelligence of multiple LLMs to unleash the potential of Multi-LLM fine-tuning.
> > >
> > > Overall, we agree that the discussion on MORL is inevitable. However, we have already implicitly compared both multi-policy and single-policy methods in our manuscript. Thank you, we will incorporate this discussion into the appendix to explicitly discuss MORL algorithms.
> > >
> > > Attached Table 1 is shown as follows:
> > >
> > > |     | Task-reward ↑ | KL divergence ↓ |
> > > |-----|---------------|-----------------|
> > > | PPO | 2.17          | 44.33           |
> > > | **GGF-PPO** | 2.24 |  38.45      |
> > > | ER-30 | 2.646       | 32.152          |
> > > | ER-17 | 2.626       | 21.73           |
> > > | ER-5 | 0.5284       | 0.3237          |
> > > | CORY | **2.668**    | **15.179**      |
> > >
> > >
> > > ### 2. **Clarification on baselines** ###
> > >
> > > More baselines are always better. However, we believe these two strong baselines, PPO and ER, should be sufficient to evaluate our method. PPO remains a strong baseline for RL fine-tuning of LLMs. Although some work has explored SAC [5], the experimental results show limited improvement comparing to PPO. In terms of preventing distribution collapse for RL fine-tuning, there is work such as NLPO [6], but its effectiveness is not as strong as ER.
> > >
> > > Thank you once again for your time. Should you have any further questions, please let us know.
> > >
> > >
> > > [1]  Hayes C F, et al. "A practical guide to multi-objective reinforcement learning and planning." AAMAS (2022)
> > >
> > > [2] Hwang M, et al. "Promptable behaviors: Personalizing multi-objective rewards from human preferences." CVPR(2024)
> > >
> > > [3] TRL: https://github.com/huggingface/trl
> > >
> > > [4] Siddique U, et al. "Learning fair policies in multi-objective (deep) reinforcement learning with average and discounted rewards." ICML(2020)
> > >
> > > [5] Wen, Muning, et al. "Entropy-Regularized Token-Level Policy Optimization for Large Language Models." CoRR (2024).
> > >
> > > [6] Ramamurthy, Rajkumar, et al. "Is Reinforcement Learning (Not) for Natural Language Processing: Benchmarks, Baselines, and Building Blocks for Natural Language Policy Optimization." ICLR (2023).

---

> > > > ### Comment · Reviewer_chZh · 2024-08-13
> > > >
> > > > I thank the authors for the discussion of MORL. I'm happy to increase my score. I would just need more information about how GGF-PPO was tuned to obtain the results in the table and if this is fair to the rest of the methods.

---

> > > > > ### Author Response · Authors · 2024-08-14
> > > > >
> > > > > Thank you for your positive feedback! Here is the information about the GGF-PPO experiment.
> > > > >
> > > > > Regarding the implementation: GGF-PPO achieves reward scalarization via weighting, enforcing constraints on the weights assigned to multiple objectives. This includes the requirement for weights to follow a strictly descending order and for their sum to equal 1. Consequently, we adjusted the weights used in PPO's gradient computation and bypassed the original handling of the KL coefficient in the TRL library.
> > > > >
> > > > > To ensure a fair comparison, we implemented the following adjustments:
> > > > > 1. We maintained the hyperparameters of GGF-PPO, except for the KL coefficient, to be consistent with those of PPO.
> > > > > 2. We ensured that the task reward and KL presented in the table represent actual measured values, rather than the weighted values utilized in PPO updates.
> > > > > 3. We conducted experiments using weight combinations of [0.9, 0.1], [0.8, 0.2], [0.7, 0.3], and [0.6, 0.4], exploring both scenarios where task reward is prioritized and where KL is prioritized, resulting in a total of eight sets of experiments. The best-performing set of results was selected for inclusion in the table.

---

### Official Review · Reviewer_F4Ro · 2024-07-12

**Soundness:** 3
**Presentation:** 4
**Contribution:** 3
**Rating:** 6
**Confidence:** 3

**Summary:**

This paper presents a Reinforcement Learning (RL) methodology to fine-tune LLMs based on Multi-Agent RL agents. One agent acts as the observer, and the other acts as the pioneer. They share knowledge through two interactions: transfer learning and role-switching. They named this methodology CORY (Coevolving with the Other You). They tested their framework with out-the-shelf LLMs and proved that learning benefits from using CORY. They compared against proximal policy optimization (PPO) in the Multi-Agent setup. Ablation studies showed the effect of model size, knowledge transfer, and role exchange. They provided theory related to RL in the context of this text-generation task and qualitative studies.

**Strengths:**

The paper is easy to follow for a person knowledgeable in RL who understands LLMs at a basic level. Their methodology provides a viable method to improve LLM fine-tuning using RL. They used modern and relevant benchmarks not only on the side of RL (PPO) but also on the side of models.

**Weaknesses:**

The Limitations section would've been a good addition to the main corpus of the paper as it addresses a big possible concern I imagined with your methodology: you require twice the amount of resources to train both agents. The average reader would enjoy this change, as some don't jump in appendices. Also, I think your claim about Moore's Law might be arguable, as current requirements for computing power are growing beyond expectation with the surge in LLMs.

**Questions:**

I think I lack intuition about how the reward signals work after the N steps. Also, I am unclear about the influence of the KL divergence and how far it is relevant not to diverge from the reference policy. Could you please give me your take on that?

**Limitations:**

The authors added a Limitations section in the appendix.

---

> ### Author Rebuttal · Authors · 2024-08-07
>
> Thank you for your positive remark and insightful feedback! We’re glad you found our method viable for improving LLM RL fine-tuning, our paper is easy to follow, and modern and relevant benchmarks are used in our experiments. Below, we provide individual responses addressing your comments.
>
> ## **R3-1** Limitation Section
> Thank you very much for your suggestion. We will carefully use the discussion regarding Moore's Law and will directly incorporate the discussion on computational power consumption into Section 6 of the revised manuscript.
>
> ## **R3-2** Reward Signal in RL Fine-Tuning
> From a reinforcement learning perspective, the N steps to generate a sentence (assuming padding is included) constitute an episode. For PPO, at the end of the N steps, gradient ascent is used to optimize the token-level policy in the direction of maximizing the return for those N steps.
>
> ## **R3-3** KL in RL Fine-Tuning
> This is a profound question. KL divergence measures the distance between the current LLM’s token-level policy and the pre-trained token-level policy. The pre-trained token-level policy is quite delicate. Deviating too much can lead to the destruction of the language modeling in the pre-trained model, that is, distribution collapse. For instance, if the probability of the eos_token in the token-level policy is too low compared to the pre-trained one, it could prevent sentences from ending, or result in the repetition of the same word until the maximum output length is reached. Distribution collapse can largely be detected by KL divergence, and experimental results also reveal a strong correlation between them.
>
> To avoid distribution collapse, current RL fine-tuning often needs to be early-stopped before reaching a catastrophic level of KL divergence. This places RL fine-tuning in a rather awkward position.
>
> Overall, the benefits of maintaining a low KL can be summarized as follows:
> 1. Stable training without collapse: This means that RL can explore more data, increasing the chances of finding better policies.
> 2. Efficient improvement: This means achieving as much performance improvement as possible with as few changes as possible to the reference policy.

---

> ### Author Response · Authors · 2024-08-12
> **Please Review Rebuttal**
>
> We kindly ask the reviewer to read and respond to our rebuttal. During the rebuttal phase, we conducted new experiments that we believe address all the concerns raised regarding the paper and may merit an increase in the score. The experiments can be summarized as follows:
>
> 1. New Baselines: Introduced two new baselines: REINFORCE and a strong baseline, Elastic Reset [1]  (see attached PDF Figure 3).
> 2. New Benchmark: All baselines have been compared on the Anthropic HH benchmark (see attached PDF Figure 4).
>
> If there are any outstanding issues, we would appreciate the opportunity to respond before the discussion period concludes. Thank you.
>
> [1] Noukhovitch, Michael, et al. "Language model alignment with elastic reset." Advances in Neural Information Processing Systems 36 (2024).

---

> > ### Comment · Reviewer_F4Ro · 2024-08-13
> >
> > First, I apologize for the late response to your rebuttal; I read it some days ago and was trying to form a better idea through the other reviews and how you addressed them. I want to thank the authors for addressing the questions and concerns in my review. I better understood how the RL part works in this NLP application, and I find it interesting how it can keep contributing to the fine-tuning tasks that are much required for LLM. I appreciate your efforts in creating new benchmarks and baselines.
> >
> > I will keep my score for the following reasons: my field of expertise is RL, and I don't feel confident giving you a higher score without a solid background in NLP. Also, my review didn't consider that you should add new baselines; for instance, after using PPO, I didn't consider it relevant to include REINFORCE, which is expected to underperform against PPO.
> >
> > I wish you all the best.

---

### Official Review · Reviewer_Syvc · 2024-07-12

**Soundness:** 3
**Presentation:** 3
**Contribution:** 3
**Rating:** 6
**Confidence:** 3

**Summary:**

This paper presents CORY, a novel approach for fine-tuning large language models (LLMs) using a sequential cooperative multi-agent reinforcement learning framework. Traditional methods, primarily based on PPO and its variants, often show suboptimal performance and risk distribution collapse. CORY addresses these issues by duplicating the LLM into two agents: a pioneer and an observer. These agents generate responses based on queries and each other’s outputs, exchanging roles periodically to foster cooperation. Experiments with GPT-2 and Llama-2 on the IMDB Review and GSM8K datasets demonstrate that CORY outperforms PPO in terms of policy optimality, resistance to distribution collapse, and training robustness. This highlights its potential as a superior methodology for refining LLMs in real-world applications.

**Strengths:**

1. The introduction of multi-agent reinforcement learning into LLM fine-tuning is both novel and compelling, with a well-justified motivation behind the designed role exchange.
2. The presentation is well-organized and clear. The comprehensive explanations and illustrative figures in section 3.2 enhance the understanding of the proposed method and its effectiveness.
3. The empirical experiments are adequate and effectively support the proposed method and its underlying rationale.

**Weaknesses:**

1. The paper lacks a theoretical analysis of the introduced framework.
2. The baseline comparison is primarily with single-agent PPO. However, there are many more advanced algorithms for LLM fine-tuning. It would be beneficial if the authors conducted additional experiments using other baseline fine-tuning algorithms.

**Questions:**

1. Can the authors provide a more detailed discussion on the relationship between emergent communication in multi-agent reinforcement learning (MARL) and the method in this paper? Since the authors mention that this work is inspired by the concept that "Languages develop through agent interactions and are shaped by societal and cultural influences," it would be beneficial to explore this connection further.
2. The paper suggests employing a collective task reward to facilitate cooperation, as outlined in Eq. 4. What would happen if the task reward is not the sum of individual task rewards? Can the authors provide empirical results or theoretical analysis to show the influence of the reward design?
3. Currently, the framework includes only two agents: the pioneer and the observer. Is it possible to introduce additional agents, and would doing so enhance performance? I hope the authors can discuss this possibility further.

**Limitations:**

Limitations have been discussed in the paper.

---

> ### Author Rebuttal · Authors · 2024-08-07
>
> Thank you for your positive remark and insightful feedback! We’re glad you found our idea is both novel and compelling, our presentation is well-organized and clear, and our experiments are adequate and effectively support the proposed method. Below, we provide individual responses addressing your comments.
>
> ## **R2-1** Theoretical Analysis
> Our paper has already provided detailed mathematical modeling for LLM RL fine-tuning (Section 2), token-level PPO (Appendix B), and CORY (Appendix C), along with an analysis from the perspective of multi-objective RL (Section 3.2). However, we are still willing to offer a theoretical analysis of why CORY works from the perspective of game theory.
>
> As shown in Figure 5 in the supplementary PDF, the training phase of CORY can be modeled as an extensive-form two-player game. This is a game of perfect but incomplete information, meaning that the observer is aware of all historical information, but the two agents do not know any specific utility functions. The relationship between the utility functions of the two LLM agents determines the type of game. In CORY, the utility function format $U=R1+R2$ corresponds to a fully cooperative game, while the format $U=R1-R2$ corresponds to a zero-sum fully competitive game.
>
> It is easy to prove that the policy combination corresponding to $U_{\text{obs}} \equiv U_{\text{pio}} \equiv 0$ is a Nash equilibrium of the zero-sum game, where $R_1 \equiv R_2$, meaning that the rewards of both agents are the same under any query. They tend to find policies that are similar in performance but not optimal during training, which is also verified by subsequent experiments (in **R2-4**).
>
> Conversely, the design of CORY cleverly utilizes the characteristics of the global optimal solution. Assuming that for any task query, there exists an answer that maximizes the task reward, it can be proven that 1. The game has Pareto optimums (constructive proof), and 2. Under the Pareto optimums, both agents receive the same and maximum rewards. However, there are two extremes among all the Pareto solutions. One is both the observer and the pioneer independently generate the optimal response (which is the true global optimum), and the other is the observer completely relies on the pioneer, merely repeating its optimal response. Thanks to the design of role-exchange and collective reward, the policies of the two agents will avoid evolving towards the second extreme and instead evolve towards the first extreme.
> Thanks, we will add this section to the appendix.
>
> ## **R2-2** Other RL Fine-tuning Baselines
> Thank you for your suggestion. We have added two baselines into our analysis: REINFORCE and Elastic Reset. Here is the results of IMDB.
> |     | Task-reward ↑ | KL divergence ↓ |
> |-----|---------------|-----------------|
> | PPO | 2.17          | 44.33           |
> |REINFORCE|0.747|56.87|
> | ER-30 | 2.65       | 32.15          |
> | ER-17 | 2.63       | 21.73           |
> | ER-5 | 0.53       | 0.32          |
> | CORY | **2.67**    | **15.18**      |
>
> The results indicate that CORY has the ability to maintain a stable balance between the two targets. The detailed training curves are shown in Figure 3 in the supplementary PDF.
>
> ## **R2-3** Emergent Communication
> Prior work has discovered that defining communication protocols as vocabularies enables agents to spontaneously emerge their own languages during cooperation. Furthermore, incorporating human labels into the learning process of communication policy can facilitate the emergence (0 to 1) of communication policy that utilize natural language [1,2].
> CORY is equivalent to a two-player cooperative game with unidirectional communication grounded in natural language. The communication policies are pre-trained LLMs. The LLM/communication policy has already undergone one emergence during pre-training. By casting the RL fine-tuning of the LLM into a multi-agent context, we anticipate a secondary emergence (1 to 100) of the LLM.
>
> [1] Lazaridou, Angeliki, Alexander Peysakhovich, and Marco Baroni. "Multi-Agent Cooperation and the Emergence of (Natural) Language." International Conference on Learning Representations. 2022.
>
> [2] Graesser, Laura, Kyunghyun Cho, and Douwe Kiela. "Emergent linguistic phenomena in multi-agent communication games." 2019 Conference on Empirical Methods in Natural Language Processing and 9th International Joint Conference on Natural Language Processing, EMNLP-IJCNLP 2019. Association for Computational Linguistics, 2019.
>
> ## **R2-4** Total Reward Design
> This is indeed an interesting question. We have noticed that Reviewer 7TR2 also raised a similar issue. Due to the space constraints of the rebuttal, please refer to **R1-1** for details on the relevant experimental setup. The results indicate that competitive scenarios significantly underperform cooperative ones. Furthermore, under complete competition (zero-sum game settings), the agents trained to a state of low KL divergence but also low performance (similar but suboptimal), which aligns with our previous game-theoretic analysis.
>
> ## **R2-5** More LLM Agents
> Yes! introducing more agents could lead to performance improvements, which represents an exciting direction for future work. We look forward to the participation of more researchers in this area. We have already developed preliminary concepts, which fundamentally concerns the design of communication topologies. There are several simple communication topologies that could be explored, including linear (chain-like), mesh (fully connected), and randomly connected. Recent research [3] has uncovered scaling laws of agent numbers in the collaboration among multiple Large Language Models (LLMs). It also discussed several other insight communication topologies for LLMs (e.g. tree-like, radial).
>
> [3] Qian, Chen, et al. "Scaling Large-Language-Model-based Multi-Agent Collaboration." arXiv preprint arXiv:2406.07155 (2024).

---

> > ### Comment · Reviewer_Syvc · 2024-08-12
> >
> > I've read the author's response and I appreciate the additional discussion and clarifications. I would like to keep my score.

---

### Official Review · Reviewer_7TR2 · 2024-07-13

**Soundness:** 2
**Presentation:** 3
**Contribution:** 2
**Rating:** 5
**Confidence:** 4

**Summary:**

In this paper, the authors study a multi-agent organization for LLM learning. Specifically, they have two LLMs, with the second one responding to the same query given the query itself and the response generated by another LLM. They author shows this method can achieve a better tradeoff of the task reward and the KL penalty.

**Strengths:**

The writing is clear and the question and proposed method is interesting.

**Weaknesses:**

Since this is an empirical paper, the expectation of the experiments, from its design to its implementation and results, would be higher. In the questions below, the reviewer mentions several points that might improve the experiments.

If the authors address the concerns about the experiments, the reviewer will consider rising their score.

**Questions:**

1. As far as the reviewer is concerned, this paper is missing several important baselines to make their experiments more convincing.

(1.1) The dual-LLM setup in the LLM debating literature.

This line of research is highly related to the paper. There, two LLMs response in turn to the previous generation of the other part, as well as to the original query. As the authors are explicitly considering a cooperative setting, the important question then is: which one is more effective, cooperation or competition, in terms of multi-agent LLM learning.

(1.2) Mixture of experts. It appears a bit unfair to compare with a single-LLM PPO baseline--after all the authors are using two LLMs.

The question is whether this pioneer-observer structure is more effective than a simpler mechanism where opinions from multiple LLMs are aggregated together by recent research in MOE.

2. Fig. 2 needs to be further explained. The major drawback is the lack of legend regarding the eta values. The reviewer is asking for the which two points (one on the CORY frontier and another on the PPO frontier) corresponding which value of eta.

A minor question: For the sub-optimal frontiers of CORY and PPO, it is unclear what is shown is the training KL and reward or that of the testing time.

**Limitations:**

The reviewer lists some questions that make them curious, expecting some clarifications. it seem that some analysis or experiments can improve the understanding of the proposed method.

(1) In the current setup, the pioneer and the observer possess the same architecture, ie, they are the same LLMs. Does this homogeneity matter? If the two LLMs are different, can the proposed method still work?

---

> ### Author Rebuttal · Authors · 2024-08-07
>
> Thank you for your detailed review and valuable feedback! We’re glad you found our proposed method interesting and the writing clear. We would like to address your concerns below.
>
> ## **R1-1** Dual-LLM Setup and Multi-LLM Learning
> There has been some work studying the collaboration of multi-LLM (including dual-LLM setup) to accomplish a shared task [1], and we will add this to our reference. To our knowledge, CORY is the first work employing a multi-agent framework for the RL fine-tuning of LLM. As highlighted by the reviewer, the dynamics of competition and cooperation among agents are crucial in multi-agent learning framework.
>
> To investigate these dynamics, we modified the original collective reward of CORY from $R_{self} + R_{other}$ to an $R_{self} + λ * R_{other}$ format. By adjusting the value and sign of $λ$, we could represent varying degrees of competition and cooperation. Additionally, to mitigate the impact of reward magnitude on training, we normalized the reward values. As demonstrated in Figure 1 in the attached PDF, across both the GSM8K and IMDB datasets, the task rewards in competitive settings were significantly lower than those in cooperative settings.
>
> Beyond experimental analyses, we believe that training multiple LLMs in a cooperative setting can be modeled as MARL problems with a shared team reward. Research in the MARL domain on credit assignment (e.g., VDN[2],QMIX[3]) could further enhance the training levels of multiple LLMs. This represents a promising direction for future development!
>
> [1] Wu, Qingyun, et al. "AutoGen: Enabling Next-Gen LLM Applications via Multi-Agent Conversation." ICLR 2024 Workshop on Large Language Model (LLM) Agents.
>
> [2] Sunehag, P., et al. "Value-Decomposition Networks For Cooperative Multi-Agent Learning Based On Team Reward." Proceedings of the 17th International Conference on Autonomous Agents and MultiAgent Systems. 2018: 2085-2087.
>
> [3] Rashid, T., et al. "QMIX: Monotonic Value Function Factorisation for Deep Multi-Agent Reinforcement Learning." International Conference on Machine Learning. PMLR, 2018: 4295-4304.
>
>
>
> ## **R1-2** MoE Baseline
> We agree that it is crucial to fairly compare the total model size as well as the number of models. Regarding the model size, our paper has already conducted ablation studies, comparing GPT2-l (770M) + CORY with GPT2-xl (1.5B) + PPO in Section 4.3, and LLaMA2 7b + CORY with LLaMA2 13b + PPO in Appendix E. The experiments demonstrate that the advantages of CORY do not come from an increase in training parameters.
>
> As for expanding the number of LLMs involved in training, we designed a simple baseline inspired by the MoE approach: aggregating the outputs of two models and selecting the output with a higher estimated value based on the Q-values. The experimental results on the IMDB dataset are as follows:
>
> |             | Task-reward ↑ | KL divergence ↓ |
> |-------------|---------------|-----------------|
> | PPO         | 2.17          | 44.33           |
> | Aggregate   | 2.36          | 47.75           |
> | **CORY**    | **2.67**     | **15.18**      |
>
> As you mentioned, we found that using MOE to aggregate responses from multiple LLMs can effectively enhance task rewards, resulting in a 9% improvement compared to PPO. However, by comparing with CORY's results, we discovered that CORY achieves a 23% improvement. This indicates that CORY provides a greater enhancement in performance because it significantly boosts the capability of individual LLM agents through collaborative training. These findings indicate that the key to the effectiveness of the pioneer-observer structure lies in facilitating interactions between models.
> ## **R1-3** Fig.2 Explanation
> We sincerely appreciate your valuable suggestion and apologize for any confusion caused. Regarding Fig. 2(c), the values of eta from left to right indeed are 1e-5, 1e-4, 1e-3, and 1e-2. We will ensure to annotate this clearly in the revised figure for better understanding.
>
> As for the sub-optimal frontiers, these actually represent the testing KL and reward, which we will explicitly clarify in the revised manuscript to avoid any ambiguity.
> ## **R1-4** Possible Heterogeneous Multi-LLM Training
> This is indeed a fascinating question. Although the paper emphasizes that CORY is a plug-and-play alternative to RL fine-tuning (requiring only a duplicate of the model to be finetuned, without the need for extra auxiliary models during training), the applicability of CORY's core ideas in heterogeneous multi-LLM finetuning is a topic worthy of exploration.
>
> To this end, we conducted experiments on the GSM8K dataset involving heterogeneous LLM training with LLaMA3-8B, LLaMA2-7B, and GPT-2. As shown in Figure 2 in the attached PDF, in the (LLaMA2-LLaMA3) combination, LLaMA2's training curve exhibits stability similar to that of original CORY, with its training KL values (consistently less than 1) significantly lower than those of single-PPO (which peak at 10). This suggests that the "knowledge transfer" mechanism can effectively alleviate the optimization pressure on LLaMA2 at the KL end. However, in the (LLaMA2-GPT2) combination, due to the insufficient capabilities of the GPT2 model, which produces many erroneous or empty responses, it fails to alleviate any training pressure on LLaMA2, resulting in a KL curve for LLaMA2 that closely mirrors the trend of single-PPO. The weaker model hinders the training of the stronger model, a similar outcome can also be observed in the (LLaMA2- LLaMA3) combination. It is evident that in a heterogeneous setting, CORY's underlying mechanisms still function, but the selection of heterogeneous LLMs and their interactions pose significant challenges, making this an issue worth further investigation.

---

> ### Author Response · Authors · 2024-08-12
> **Please Review Rebuttal**
>
> We kindly ask the reviewer to read and respond to our rebuttal. During the rebuttal phase, we conducted new experiments that we believe address all the concerns raised regarding the paper and may merit an increase in the score. The experiments can be summarized as follows:
>
> 1. Dual-LLM Setup: Evaluated both cooperative and competitive settings (see attached PDF Figure 1).
> 2. Heterogeneous Setup: Implemented two different LLMs in CORY to assess the effectiveness of the proposed mechanism (see attached PDF Figure 2).
> 3. New Baselines: Introduced two new baselines: REINFORCE and a strong baseline, Elastic Reset [1]  (see attached PDF Figure 3).
> 4. New Benchmark: All baselines have been compared on the Anthropic HH benchmark (see attached PDF Figure 4).
>
> If there are any outstanding issues, we would appreciate the opportunity to respond before the discussion period concludes. Thank you.
>
> [1] Noukhovitch, Michael, et al. "Language model alignment with elastic reset." Advances in Neural Information Processing Systems 36 (2024).

---

> > ### Comment · Reviewer_7TR2 · 2024-08-12
> > **Thanks for your response**
> >
> > The authors' response has addressed my major concern. I have increased my score accordingly.

---

### Author Rebuttal · Authors · 2024-08-07

We would like to express our gratitude to all the reviewers for their contributions and insightful comments. We appreciate that all the reviewers find our writing and presentation is clear and well-organized.
We are encouraged by the reviewers' appreciation that our idea is novel and interesting (Syvc, 7TR2), that our method is viable and possibly influence other work to build on it (F4Ro, chZh), that the analysis from the multi-objective RL is comprehensive and interesting (Syvc, chZh), that the experiments using modern and relevant benchmarks are adequate (Syvc, F4Ro), and includes enough details about the experiments (chZh).

To help address the concerns of the reviewers and facilitate further discussion, we are attaching a PDF with five figures that we reference in each of our reviewer-specific rebuttals. The figures are:
**Figure 1**: Training curves of cooperative and competitive settings between two LLMs on IMDB and GSM8K datasets. This is mostly in response to Reviewer 7TR2 and Reviewer Syvc.
**Figure 2**: Training curves of heterogeneous LLM settings under the CORY framework on GSM8K dataset. This is mostly in response to Reviewer 7TR2.
**Figure 3**: Comparisons to REINFORCE and Elastic Reset on IMDB and GSM8K datasets. This is mostly in response to Reviewer Syvc and Reviewer chZh.
**Figure 4**: Training curves on Anthropic-HH. This is mostly in response to Reviewer chZh.

**Figure 5**: Game-theoretic modeling of CORY. This is mostly in response to Reviewer Syvc.

---

### Comment · Area_Chair_zTTL · 2024-08-12
**Discussion period**

The discussion period is almost over, so both authors and reviewers please respond to any unaddressed questions. Reviewers, be sure that you have all of the information you need from the authors, since after the 13th, they won't be able to respond.

---

### Author Response · Authors · 2024-08-14

We thank all the reviewers for their valuable suggestions and active engagement during the discussion phase. Your time and effort have clarified our work, and we sincerely appreciate the constructive feedback.

---

### Decision · Program_Chairs · 2024-09-25

**Decision:**

Accept (poster)

**Comment:**

While there were some significant ethical concerns to walk through (e.g. misinformation creation) and some concern over the lack of evaluation and RL algorithm diversity, these were largely resolved during the rebuttal process. The expanded limitation section addresses the ethical concerns very thoughtfully, and I believe PPO is sufficient for the purposes of this paper.